# Feature-Based Belief Aggregation for Partially Observable Markov Decision Problems

## Abstract

We consider a finite-state partially observable Markov decision problem (POMDP) with an infinite horizon and a discounted cost, and we propose a new method for computing a cost function approximation that is based on features and aggregation. In particular, using the classical belief-space formulation, we construct a related Markov decision problem (MDP) by first aggregating the unobservable states into feature states, and then introducing representative beliefs over these feature states. This two-stage aggregation approach facilitates the use of dynamic programming methods for solving the aggregate problem and provides additional design flexibility. The optimal cost function of the aggregate problem can in turn be used within an on-line approximation in value space scheme for the original POMDP. We derive a new bound on the approximation error of our scheme. In addition, we establish conditions under which the cost function approximation provides a lower bound for the optimal cost. Finally, we present a biased aggregation approach, which leverages an optimal cost function estimate to improve the quality of the approximation error of the aggregate problem.

## 1 Introduction

We consider an infinite horizon partially observable Markov decision problem (POMDP) with a discounted cost criterion. In particular, we assume that the state space $X$ consists of $n$ states, denoted by $1, \ldots, n$. The control space $U$ and observation space $Z$ are finite, with elements denoted by $u$ and $z$, respectively. Transitions from state $i$ to state $j$ under control $u$ occur at discrete times $k$ according to transition probabilities $p_{ij}(u)$. Each transition produces an observation $z$ with probability $p(z \mid j, u)$ and incurs a cost of $g(i, u, j)$. At each stage $k$, we select a control $u_k \in U$ based on the history of observations $z_1, \ldots, z_{k-1}$ and controls $u_0, \ldots, u_{k-1}$. The objective is to minimize the expected value of the total cost, discounted with a discount factor $\alpha \in (0, 1)$.

It is well known that a POMDP admits an equivalent perfect-state information formulation, leading to a Markov decision problem (MDP); see e.g., Åström (1965). This formulation involves the belief state $b = \big(b(1), \ldots, b(n)\big)$, where each component $b(i)$ represents the conditional probability that the system is in state $i$, given the history of controls and observations. The belief state is updated recursively through a belief estimator:

$$b' = F(b, u, z), \tag{1}$$

where $F$ is a given function, and $b'$ denotes the updated belief. The set of belief states is called the belief space and is denoted by $B$.

We consider policies $\mu$ that map the belief space $B$ to the control space $U$. The cost function of a policy $\mu$, denoted by $J_\mu$, maps $B$ to the real line $\Re$, and is defined at any initial belief state $b_0 \in B$ as

$$J_\mu(b_0) = \lim_{N \to \infty} \mathop{E}_{\substack{z_k \\ k=1,\ldots,N-1}} \left\{ \sum_{k=0}^{N-1} \alpha^k \hat{g}\big(b_k, \mu(b_k)\big) \right\}, \tag{2}$$

where $E\{\cdot\}$ denotes expected value, and the stage cost $\hat{g}(b,u)$ and the conditional distribution $\hat{p}(z\,|\,b,u)$ of $z$ given $b$ and $u$ are defined by

$$\hat{g}(b,u) = \sum_{i=1}^{n} b(i) \sum_{j=1}^{n} p_{ij}(u) g(i,u,j),$$

$$\hat{p}(z\,|\,b,u) = \sum_{i=1}^{n} b(i) \sum_{j=1}^{n} p_{ij}(u) p(z\,|\,j,u). \tag{3}$$

The optimal cost function $J^*$, derived by optimizing over all possible policies $\mu$, uniquely satisfies the Bellman equation

$$J^*(b) = \min_{u \in U} \left[ \hat{g}(b,u) + \alpha \sum_{z \in Z} \hat{p}(z\,|\,b,u) J^*\big(F(b,u,z)\big) \right], \qquad \text{for all } b \in B, \tag{4}$$

see, e.g., (Bertsekas, 2017, Section 3.5.2) and (Bertsekas, 2012, Section 5.6), or (Krishnamurthy, 2016, Thm. 7.6.2). Once the optimal cost function $J^*$ is computed, an optimal policy $\mu^*$ that satisfies $J_{\mu^*}(b) = J^*(b)$ for all $b \in B$ can be determined by minimizing the right-hand side of Eq. (4), i.e.,

$$\mu^*(b) \in \arg\min_{u \in U} \left[ \hat{g}(b,u) + \alpha \sum_{z \in Z} \hat{p}(z\,|\,b,u) J^*\big(F(b,u,z)\big) \right], \qquad \text{for all } b \in B;$$

see, e.g., (Bertsekas, 2012, Section 1.2).

For many POMDP of practical interest, neither the optimal cost function $J^*$ nor the optimal policy $\mu^*$ can be computed exactly. Consequently, there has been extensive research on approximation schemes. Within the literature on POMDP approximations, we note three principal approaches: value-based approximation methods Pineau et al. (2006); Kurniawati et al. (2008); Ong et al. (2010); Lovejoy (1991); Littman et al. (1995); Zhou & Hansen (2001); Poupart & Boutilier (2002); Yu & Bertsekas (2004); Roy et al. (2005); Saldi et al. (2017), heuristic search Smith & Simmons (2004); Silver & Veness (2010); Somani et al. (2013); Sunberg & Kochenderfer (2018); Wu et al. (2021), and policy-based approximation methods Schulman et al. (2017); Cobbe et al. (2021); Ni et al. (2022). The first two approaches approximate the optimal cost function $J^*$ by using off-line and/or on-line computations, whereas policy-based methods parameterize the policy and optimize it via gradient-based algorithms.

In this paper, we focus on the aggregation approach, a value-based approximation methodology for MDP with large state spaces. The conceptual aggregation framework of the present paper was first formulated in (Bertsekas, 2012, Section 6.5), and its application to POMDP was given in (Bertsekas, 2012, Example 6.5.5). In this framework, we approximate a given MDP with a reduced-space MDP (called *aggregate MDP*), which can be designed by using domain-specific knowledge. The aggregate MDP can be solved by using dynamic programming (DP) methods, among other techniques.

The most straightforward approach to apply aggregation to the belief-space formulation of a POMDP, is to select a finite set of representative beliefs within the belief space and form an MDP with a finite state space. However, this approach faces practical challenges when dealing with a large number of unobservable states, i.e., when $n$ is large. In particular, without favorable structural properties, solving the aggregated MDP involves updating belief states according to Eq. (1). This computation can become prohibitively expensive for large values of $n$, even if the transition probabilities $p_{ij}(u)$ are sparse; i.e., for every $i$ and $u$, the number of $j$ such that $p_{ij}(u) > 0$ is small relative to $n$. Furthermore, the effective selection of representative beliefs from the high-dimensional belief space $B$ is challenging, and the quality of the selection significantly impacts the accuracy of the resulting cost function approximation.

To address these difficulties, we introduce a new belief aggregation method, which is better suited to the POMDP structure. Rather than directly aggregating beliefs, *we aggregate the unobservable states into suitably designed feature states and then define representative beliefs over these feature states.* The choice of feature states can leverage problem-specific insights, providing an additional layer of flexibility, and leading to a more informed and effective selection of representative beliefs for constructing the associated MDP.

Moreover, because the number of feature states is typically much smaller than $n$, updating beliefs over these feature states becomes computationally tractable when the transition probabilities $p_{ij}(u)$ are sparse.

We note that feature-based aggregation techniques have been proposed for MDP with perfect-state information; see, e.g., Tsitsiklis and Van Roy (Tsitsiklis & van Roy, 1996), Bertsekas and Tsitsiklis (Bertsekas & Tsitsiklis, 1996, Section 3.1.2), and Bertsekas (Bertsekas, 2012, Section 6.5), (Bertsekas, 2018). In these methods, features of the (observed) states are used to guide the design of the aggregated MDP. When applied to POMDP, this means constructing features of the *beliefs over the unobservable states*, while the dimension of the belief vector remains $n$. In contrast, our method constructs *features of the unobservable states*, resulting in belief vectors defined over the feature space, whose dimension can be significantly smaller than $n$.

Similar to other aggregation methods, the aggregate MDP obtained through our method can be solved either using explicit problem data or through simulation-based estimates. In particular, when the transition or observation probabilities are not available in closed form, the quantities required to construct and solve the aggregate MDP can be estimated through sampling from a simulator. This flexibility allows the proposed framework to be applied in both model-based and simulation-based contexts, and to remain effective even when only sample trajectories are accessible.

As part of our analysis, we provide a bound for the error between the cost of the aggregate problem and the optimal cost. Our result improves the existing error bound of Tsitsiklis and Van Roy (Tsitsiklis & van Roy, 1996), as we relax certain restrictions on how the aggregated MDP is constructed. Actually, our error bound analysis does not rely on properties specific to POMDP. As a result, it can be applied to aggregation methods for MDP with only minor modifications. A generalized version of this analysis, tailored specifically to MDPs, is presented in Anonymous (2025b).

Apart from introducing the new aggregation framework and its theoretical analysis, we also provide an extension of our method that leverages an existing cost function approximation $V$. This extension is based on the *biased aggregation* framework introduced in Bertsekas (2019a); see also (Bertsekas, 2019b, Section 6.5). In particular, given an MDP and the function $V$, biased aggregation formulates a modified MDP, whereby the stage cost of the original MDP is altered using $V$. We demonstrate on our running example that the feature-based belief aggregation framework for POMDP can incorporate a bias function, with significant performance improvement. We believe this is the first experimental demonstration of the benefits obtained from the biased aggregation approach.

In summary, our contributions are the following:

(i) We introduce a feature-based aggregation scheme that generalizes existing aggregation methods for POMDP and facilitates DP calculations for solving the resulting aggregated problems.

(ii) We establish a new bound on the approximation error of the cost function obtained through aggregation. This bound extends existing results by relaxing certain restrictions on how the aggregated MDP are constructed.

(iii) We present an extension of our method that uses an existing cost function approximation as a bias function. This extension offers enhanced flexibility and substantial computational savings, while preserving theoretical guarantees.

To facilitate our discussion of the aggregation method and illustrate the theoretical results, we introduce the following POMDP, adapted from (Bertsekas, 2012, Example 1.3.1). This POMDP will serve as a running example in the next two sections. Owing to its favorable structure, both the optimal cost function $J^*$ and the optimal policies can be computed conveniently. This allows us to compare the cost function approximations obtained via aggregation with the optimal costs.

**Example 1.1** (Treasure Hunting). *Consider a sequential search problem involving $N$ sites. Each site $\ell$ may contain a treasure with value $v_\ell$. Searching site $\ell$ incurs a cost $c_\ell$ and reveals the treasure with probability $\beta_\ell$ (assuming a treasure is present). At each stage, we may choose a site to search or terminate the search.*

The objective is to design a search policy that minimizes the expected infinite-horizon search cost, discounted by a given factor $\alpha \in (0, 1)$.

We model this problem as a POMDP. The unobservable state space $X$ consists of $2^N + 1$ states, representing all possible combinations of treasure presence across the $N$ sites, along with a termination state. In particular, a non-terminated state $i$ can be represented as $i = (i^1, i^2, \ldots, i^N)$, with $i^\ell \in \{0, 1\}$, $\ell = 1, 2, \ldots, N$. Here $i^\ell = 1$ means that the $\ell$th site contains a treasure and $i^\ell = 0$ means it does not. As a result, the state space $X$ can be defined as

$$X = \{t\} \cup \left\{ (i^1, i^2, \ldots, i^N) \,|\, i^\ell \in \{0, 1\}, \, \ell = 1, 2, \ldots, N \right\},$$

where $t$ represents the termination state. The control space $U$ contains $N + 1$ options: searching one of the $N$ sites or terminating the search. At a given time period, if a site $\ell$ that contains a treasure is searched, the search is successful with probability $\beta_\ell$. In this case, the cost of the search is $c_\ell - v_\ell$, and the state transitions to the state where the treasure is removed from site $\ell$, while the status of the other sites remains unchanged. The search is unsuccessful with the complementary probability $1 - \beta_\ell$. In this case, the cost of the search is $c_\ell$, and the state remains unchanged. At the end of a search, the belief state is suitably updated (using Bayes' rule in the case of an unsuccessful search; see (Bertsekas, 2012, Example 1.3.1)).

Here the belief state $b$ is a $(2^N + 1)$-dimensional vector, and we assume that the initial belief state is given. The belief update function $F$ from Eq. (1), the stage cost $\hat{g}$ and the observation probability $\hat{p}$ in Eq. (3) can be defined accordingly.[1] In the following sections, we use this example to illustrate our aggregation method and to illustrate the approximation error of our method.

The paper is organized as follows. In Section 2, we introduce the feature-based belief aggregation method, including the dynamic system constructed through aggregation and the resulting optimal control problem. In Section 3, we analyze the error of the cost function approximation obtained via aggregation, relative to the optimal cost. In Section 4, we present computational results that support our analysis. In Section 5, we describe the biased aggregation approach and we provide favorable computational results.

## 2 Feature-Based Belief Aggregation for POMDP

In this section, we present our aggregation method. We introduce a dynamic belief system and we formulate the aggregate problem, following the framework of (Bertsekas, 2012, Section 6.5). This problem takes the form of an MDP whose state space consists of selected representative beliefs over the feature states. Finally, we describe computational approaches for solving the aggregate problem and discuss the computational savings of our methodology.

### 2.1 Dynamic Belief System

The first step of our aggregation method is constructing a finite set $\mathcal{F}$, which we call *feature space*. Its elements, generally denoted by $x$ and $y$, are called *feature states*. We connect feature states with the original (unobservable) system states as follows:

(a) With every feature state $x$, we associate a subset $I_x \subset \{1, 2, \ldots, n\}$. We require that the sets $I_x$, $x \in \mathcal{F}$, are disjoint.

(b) With every feature state $x$, we associate its *disaggregation probability distribution* $\{d_{xi} \,|\, i = 1, 2, \ldots, n\}$. We require that

$$d_{xi} = 0, \qquad \text{for all } i \notin I_x. \tag{5}$$

(c) With every state $j = 1, 2, \ldots, n$, we associate its *aggregation probability distribution* $\{\phi_{jy} \,|\, y \in \mathcal{F}\}$. We require that

$$\phi_{jy} = 1, \qquad \text{for all } j \in I_y, \, y \in \mathcal{F}. \tag{6}$$

---

[1]Due to the favorable structure of the problem, the belief vector can be compactly represented with only $N + 1$ dimensions. The optimal policy can be derived using this compact representation; further details can be found in (Bertsekas, 2012, Section 1.3).

The disaggregation and aggregation probability distributions of Eqs. (5) and (6) specify a controlled dynamic system involving both the original (unobservable) system states $i, j$ and the (unobservable) feature states $x, y$. The transition diagram of this dynamic system is given as Fig. 1.

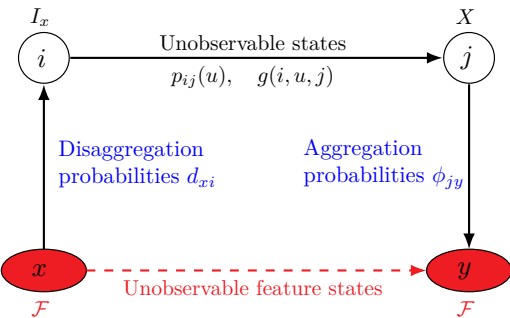

Figure 1: The controlled dynamic feature-state system constructed through the disaggregation and aggregation probability distributions in Eqs. (5) and (6).

The feature states $x$ and associated sets $I_x$, along with the aggregation and disaggregation probabilities, can be designed using problem-specific structure and engineering intuition. Alternatively, suitable feature states can be derived automatically through deep learning techniques. For instance, given a simulator of the POMDP and a policy, one can generate a dataset of belief–cost pairs and train a deep neural network to approximate the policy's cost function. The learned representation can then serve as a basis for feature-state system design; see Bertsekas (2018) for a detailed discussion for perfect-state-information problems. In this paper, we do not prescribe a specific way of constructing feature states $x$ and associated sets $I_x$ in our method, as the appropriate approach depends on the problem domain. Instead, we allow arbitrary choices of feature states (as long as the sets $I_x$ are disjoint), consistent with the classical aggregation theory framework.

We illustrate the process of designing feature states using our running example, as described below.

**Example 2.1** (Features for Treasure Hunting). *Recall that in Example 1.1, each nonterminated state $i$ can be represented by an $N$-dimensional vector $(i^1, i^2, \ldots, i^N)$ with $i^\ell \in \{0, 1\}$, $\ell = 1, 2, \ldots, N$, indicating the presence of a treasure in site $\ell$. We may define as a feature state $x$ the index of the site with the largest undiscovered value, with ties broken consistently using a fixed rule. In particular, for every state $i = (i^1, i^2, \ldots, i^N)$ such that $i^\ell = 1$ for some $\ell$, we define its feature state $x$ as*

$$x \in \arg \max_{\ell=1,\ldots,N;\, i^\ell=1} v_\ell. \tag{7}$$

*In addition, we assign feature state $x = 0$ to the states $i = (0, \ldots, 0)$ (nonterminated state with no remaining treasures) and $t$ (terminated state). Thus the feature space is $\mathcal{F} = \{0, 1, \ldots, N\}$. The sets $I_x$, $x = 0, 1, \ldots, N$, form a partition of the state space $X$. The disaggregation probabilities $d_{xi}$ could be a uniform distribution over all states $i$ in $I_x$, while the aggregation probabilities $\phi_{ix}$ map each state $i$ deterministically to the corresponding feature state $x$ via Eq. (7).*

*For an illustration, let us assume that there are two sites with the treasure of the first site larger than the one of the second site, i.e., $v_1 > v_2$. The state and feature spaces are*

$$X = \{(0,0), (1,0), (0,1), (1,1), t\}, \qquad \mathcal{F} = \{0, 1, 2\}.$$

*The sets $I_x$, $x = 0, 1, 2$, are given by*

$$I_0 = \{(0,0), t\}, \qquad I_1 = \{(1,0), (1,1)\}, \qquad I_2 = \{(0,1)\}.$$

*Because the treasure value at the first site is larger than that of the second site, the feature state associated with the state $(1, 1)$ is 1. As indicated above, this choice of feature states leads to a feature space of size $N + 1$, which is much smaller than $2^N + 1$ when $N$ is large. We will use this definition of $X$ and $\mathcal{F}$ in our subsequent discussion of the running example.*

*Alternatively, we can define the feature space $\mathcal{F}$ by grouping several search sites as one. For instance, we can partition the $N$ sites into $L$ contiguous groups of equal size and define a feature state as $x = (x^1, x^2, \ldots, x^L)$, where each component $x^l$ is $1$ if any site within the $l$th group contains a treasure, and $0$ otherwise. This type of aggregation leads to a feature space of size $2^L + 1$.*

Having defined connections between feature states and states, we now introduce the associated beliefs. We denote by $Q$ the belief space over $\mathcal{F}$, i.e., the set of probability distributions over the features in $\mathcal{F}$. Elements of $Q$ are referred to as *feature beliefs* and are denoted by $q = \{q(x) \,|\, x \in \mathcal{F}\}$. We aggregate this set into a finite subset $\tilde{Q}$ that consists of $m$ elements of $Q$, which we refer to as *representative feature beliefs* $\tilde{q} = \{\tilde{q}(x) \,|\, x \in \mathcal{F}\}$; see Fig. 2. Our framework places no restriction on how the set $\tilde{Q}$ is constructed, allowing any discretization of the feature-belief space that is convenient for the problem at hand. These representative feature beliefs are associated with feature beliefs as follows.

(d) For each feature belief $q \in Q$, we define a probability distribution over $\tilde{Q}$, denoted by $\{\psi_{q\tilde{q}} \,|\, \tilde{q} \in \tilde{Q}\}$. We refer to this distribution as the *belief aggregation probability of $q$*. We require that

$$\psi_{\tilde{q}\tilde{q}} = 1, \qquad \text{for all } \tilde{q} \in \tilde{Q}. \tag{8}$$

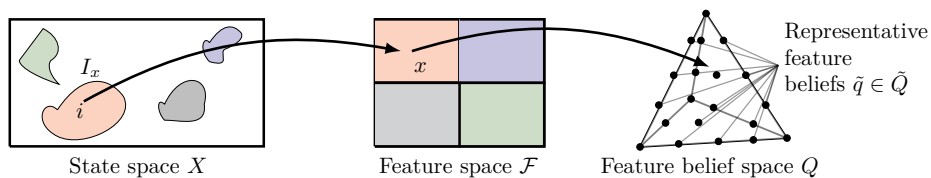

State space $X$       Feature space $\mathcal{F}$       Feature belief space $Q$

Figure 2: Our feature-based belief aggregation method: (i) we map the unobservable states $i$ into the feature states $x$; (ii) we aggregate beliefs over the feature states $q$ into a finite set of representative feature beliefs $\tilde{q}$. In this illustration, we aggregate some of the $n$ states $i$ into 4 different feature states $x$; i.e., $\mathcal{F}$ contains 4 elements. The set $I_x$ and its corresponding feature $x$ are shown in the same color. Then we choose some representative feature beliefs from the feature belief space $Q$ (i.e., the 3-dimensional unit-simplex).

The feature space $\mathcal{F}$ and the aggregation probabilities can be designed based on engineering intuition or learned from data; Bertsekas (2018). As illustrated in Fig. 2, these features often lead to "irregular shapes" of the sets $I_x$. This is a generic characteristic of feature-based aggregation schemes, which helps to capture the "nonlinearity" of the optimal cost function Bertsekas (2018).

In what follows, we use our running example to demonstrate one special choice of the set $\tilde{Q}$ and its associated belief aggregation probabilities.

**Example 2.2.** *A general way to construct the set of representative feature beliefs $\tilde{Q}$ is through a uniform discretization of $Q$. To this end, we define the set $\tilde{Q}$ as*

$$\tilde{Q} = \left\{ \tilde{q} \,\middle|\, \tilde{q} \in Q, \tilde{q}(x) = \delta_x/\rho, \sum_{x \in \mathcal{F}} \delta_x = \rho, \delta_x \in \{0, \ldots, \rho\} \right\}, \tag{9}$$

*where $\rho$ is a design parameter, which can be interpreted as the* discretization resolution. *For our two-site search problem, cf. Example 2.1, the set $Q$ is a 2-dimensional simplex and $\tilde{Q}$ is a finite subset of it.*

*For every feature belief $q \in Q$, we define its belief aggregation probability as*

$$\psi_{q\tilde{q}} = 1 \text{ if and only if } \tilde{q} \in \arg\min_{\tilde{q}' \in \tilde{Q}} \|q - \tilde{q}'\|, \tag{10}$$

*where ties in the* $\arg\min$ *are broken according to some rule, and $\|\cdot\|$ denotes the maximum norm. In the literature, aggregation schemes where for every $q \in Q$, $\psi_{q\tilde{q}} = 1$ for some $\tilde{q} \in \tilde{Q}$ are referred to as* hard aggregation. *In the subsequent discussion, we will use hard aggregation based on the representative feature beliefs $\tilde{Q}$ and belief aggregation probabilities $\psi_{q\tilde{q}}$ defined in Eqs. (9) and (10), respectively, to demonstrate the cost function approximations computed via our scheme.*

Consistent with the standard aggregation framework (see (Bertsekas, 2012, Section 6.5) or (Bertsekas, 2019b, Section 6.2)), the disaggregation and aggregation probabilities specify a controlled dynamic system involving beliefs $b$, feature beliefs $q$, and representative feature beliefs $\tilde{q}$. This system includes four types of transitions, as described below.

(i) From a representative belief $\tilde{q}$, we apply the disaggregation probabilities $d_{xi}$ and construct a belief state $b$ according to

$$b(i) = \sum_{x \in \mathcal{F}} \tilde{q}(x) d_{xi}, \qquad i = 1, 2, \ldots, n. \tag{11}$$

(ii) From the belief $b$, we apply a control $u \in U$, which generates an observation $z$ according to $\hat{p}(z \mid b, u)$ and incurs the cost $\hat{g}(b, u)$. We then update the belief as $b' = F(b, u, z)$.

(iii) From the updated belief $b'$, we generate the feature belief

$$q'(y) = \sum_{j=1}^{n} b'(j) \phi_{jy}, \qquad \text{for all } y \in \mathcal{F}. \tag{12}$$

(iv) From the feature belief $q'$, we generate a new representative feature belief $\tilde{q}'$ according to the belief aggregation probability $\psi_{q'\tilde{q}'}$.

To express these transitions succinctly, we define a mapping $D : \tilde{Q} \mapsto B$ by

$$\big(b(1), b(2), \ldots, b(n)\big) = D(\tilde{q}), \qquad b(i) \text{ satisfy Eq. (11) for all } i. \tag{13}$$

We denote the image of this mapping as $\tilde{B}$, i.e.,[2]

$$\tilde{B} = \big\{ b \in B \mid b = D(\tilde{q}) \text{ for some } \tilde{q} \in \tilde{Q} \big\}. \tag{14}$$

In addition, we define a mapping $\Phi : B \mapsto Q$ by

$$q' = \Phi(b'), \qquad q'(x) \text{ satisfies Eq. (12) for all } x \in \mathcal{F}. \tag{15}$$

The resulting aggregate controlled dynamic belief system is illustrated in Fig. 3.

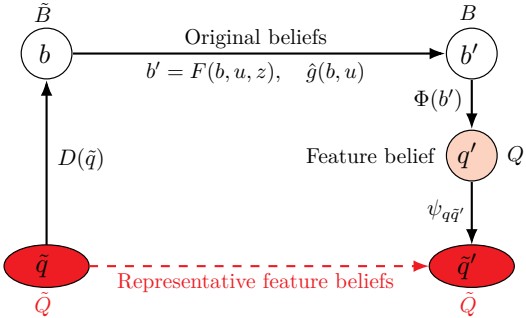

Figure 3: The controlled dynamic belief system constructed through our feature-based belief aggregation method.

Note that our framework encompasses existing aggregation methods as special cases. In particular, when each state $i$ can be identified exactly from the observation, the formulation reduces to the aggregation framework for perfect state information problems described in (Bertsekas, 2012, Section 6.5). Conversely, by setting $\mathcal{F} = X$, we recover the aggregation method applied directly to the belief space, as discussed in (Bertsekas, 2012, Example 6.5.5). Even within these contexts, our framework goes further by allowing more flexible choices of aggregation and disaggregation probabilities while maintaining theoretical guarantees, as will be discussed in Section 3.2. In addition, our framework addresses the special challenges posed by large-scale POMDP problems, for which there are few (if any) theoretical or computational studies in the context of aggregation (and no studies in the case of biased aggregation, as we will discuss in Section 5).

---

[2] We will show in Section 3 that the representative feature beliefs $\tilde{q}$ acts as "doubles" of the beliefs in $\tilde{B}$.

### 2.2 The Aggregate Problem

Having defined the dynamic belief system through the procedure illustrated in Fig. 3, we obtain an optimal control problem with state space $\tilde{Q}$ and control space $U$. A cost function for this problem is a vector in $\Re^m$, where $m$ is the number of elements in $\tilde{Q}$. We denote such a vector by $r$, where $r_{\tilde{q}}$ is the component corresponding to $\tilde{q} \in \tilde{Q}$. The associated Bellman equation is characterized by an operator $H$, which maps $r \in \Re^m$ to $Hr$ with components $(Hr)(\tilde{q})$ given by

$$(Hr)(\tilde{q}) = \min_{u \in U} \left[ \hat{g}\big(D(\tilde{q}), u\big) + \alpha \sum_{z \in Z} \hat{p}\big(z \mid D(\tilde{q}), u\big) \sum_{\tilde{q}' \in \tilde{Q}} \psi_{G(\tilde{q}, u, z)\tilde{q}'} r_{\tilde{q}'} \right], \quad \tilde{q} \in \tilde{Q}, \tag{16}$$

where the mapping $G : \tilde{Q} \times U \times Z \mapsto Q$ is defined as

$$G(\tilde{q}, u, z) = \Phi\Big(F\big(D(\tilde{q}), u, z\big)\Big). \tag{17}$$

The function $G$ compactly represents a series of operations. Given a representative belief $\tilde{q} \in \tilde{Q}$, we obtain its corresponding belief state $D(\tilde{q})$ through disaggregation. After applying a control $u$ and receiving an observation $z$, we obtain an updated belief state $F\big(D(\tilde{q}), u, z\big)$. The updated belief state is mapped to a feature belief state through a two-step aggregation process (state belief $b'$ to feature belief $q'$ to representative belief $\tilde{q}'$); cf. Fig. 3.

For the operator $H$, we have the following result.

**Proposition 1.** *Let $r$ and $r'$ be vectors in $\Re^m$ and $\| \cdot \|$ be the maximum norm.*

 *(a) If $r \le r'$, then $Hr \le Hr'$.*

 *(b) We have $\|Hr - Hr'\| \le \alpha \|r - r'\|$.*

The proof of this proposition can be obtained by straightforward modifications of standard arguments; see, e.g., (Bertsekas, 2019b, Section 6.2). It relies on the fact that both $\hat{p}$ and $\psi$ are probability distributions. In what follows, we will refer to the properties of $H$ stated in parts (a) and (b) of Prop. 1 as *monotonicity* and *contractivity*, respectively.

A direct consequence of Prop. 1 is that the operator $H$ has a fixed point $r^* \in \Re^m$, which satisfies uniquely the equation $r^* = Hr^*$ and is the optimal cost vector of the aggregate problem. This vector allows us to approximate the optimal cost function of the original POMDP [cf. Eq. (4)] via the following interpolation formula:

$$\tilde{J}(b) = \sum_{\tilde{q} \in \tilde{Q}} \psi_{\Phi(b)\tilde{q}} r_{\tilde{q}}^*; \tag{18}$$

see Fig. 4 for an illustration.

Given this cost function approximation, we can compute a suboptimal policy via one-step lookahead minimization

$$\tilde{\mu}(b) \in \arg\min_{u \in U} \left[ \hat{g}(b, u) + \alpha \sum_{z \in Z} \hat{p}(z \mid b, u) \tilde{J}(F(b, u, z)) \right]. \tag{19}$$

The quality of this policy depends on the difference between the cost function approximation $\tilde{J}$ [cf. Eq. (18)] and the optimal cost function $J^*$ [cf. Eq. (4)], which we refer to as the *approximation error*, and analyze in Section 3.

### 2.3 Methods for Solving the Aggregate Problem

The aggregate problem is an MDP with a finite state space (the set $\tilde{Q}$ of representative feature beliefs) and the finite control space $U$. This MDP can be solved with standard DP methods. In particular, we can use the exact value iteration (VI) algorithm to compute the optimal solution $r^*$. This is the fixed-point iteration

$$r^{k+1} = Hr^k, \tag{20}$$

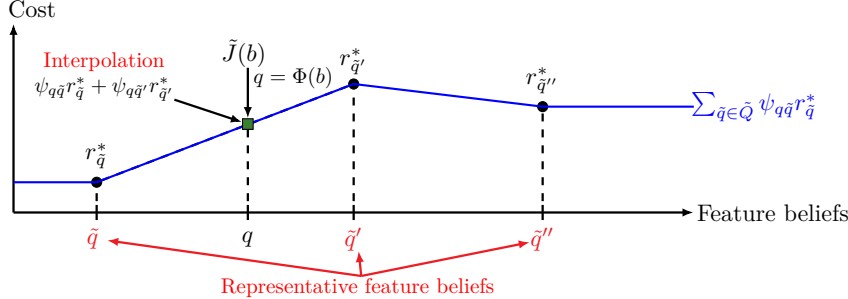

Figure 4: Illustration of the interpolation formula given by Eq. (18). It is used to obtain the cost function approximation $\tilde{J}$. The illustration is based on an approximation with three representative feature beliefs: $\tilde{Q} = \{\tilde{q}, \tilde{q}', \tilde{q}''\}$.

that starts from some initial guess $r^0$, where $H$ is the contraction mapping of Eq. (16).

An asynchronous version of the algorithm can also be applied to solve this problem. In such an algorithm, at each iteration $k$, we apply $H$ to a single component of the current representative belief vector $\tilde{q}_k$, and leave all other components unchanged. In particular, for a sequence of representative feature beliefs $\{\tilde{q}_0, \tilde{q}_1, \dots\}$, given $r^k$ and $\tilde{q}_k$, we first compute the vector $b_k$ by

$$b_k(i) = \sum_{x \in \mathcal{F}} \tilde{q}_k(x) d_{xi}, \qquad i = 1, 2, \dots, n; \tag{21}$$

cf. Eq. (11). We then update only the component $r^{k+1}_{\tilde{q}_k}$ according to

$$r^{k+1}_{\tilde{q}_k} = \min_{u \in U} \left[ \hat{g}(b_k, u) + \alpha \sum_{z \in Z} \hat{p}(z \mid b_k, u) \sum_{\tilde{q}' \in \tilde{Q}} \psi_{\Phi\left( \left( F(b_k, u, z) \right)_{\tilde{q}'} \right)} r_{\tilde{q}'} \right], \tag{22}$$

while keeping all other components unchanged:

$$r^{k+1}_{\tilde{q}} = r^k_{\tilde{q}}, \qquad \text{if } \tilde{q} \neq \tilde{q}_k. \tag{23}$$

The justification and convergence properties of this method parallel those of the standard asynchronous VI algorithm; see Bertsekas (1982; 1983).

For both the exact and asynchronous VI algorithms, the vector $b_k$ and the functions $\hat{g}$, $\hat{p}$, and $F$ can be written explicitly in closed form. Their values may also be estimated through simulation methods; e.g., employing particle filtering to approximate the belief estimator $F$. Hence, the same algorithmic structure applies whether explicit problem data or a simulator is available. In either case, regardless of the VI version or computation method, our aggregation scheme results in significant computational benefits. We illustrate these benefits by considering specifically the asynchronous VI given in Eqs. (21)–(23), under the assumption that $b_k$ and the functions $\hat{g}$, $\hat{p}$, and $F$ are evaluated explicitly in closed form.

The first benefit is that storing and retrieving low-dimensional representative feature beliefs $\tilde{q}$ is simpler than handling full belief states $b$. The second benefit is that the computational effort required by Eqs. (21)–(23) is greatly reduced when the set $\cup_{x \in \mathcal{F}} I_x$ contains far fewer elements than $n$, and/or when the transition probabilities $p_{ij}(u)$ are sparse for all $i \in \cup_{x \in \mathcal{F}} I_x$ and $u \in U$. Indeed, computing $b_k$ via Eq. (21) involves significantly fewer unobservable states $i$ when the set $\cup_{x \in \mathcal{F}} I_x$ is small. Moreover, sparse transition probabilities $p_{ij}(u)$, together with the small number of nonzero components in $b_k$, substantially decrease the number of unobservable states $i, j$ needed to evaluate $\hat{g}(b_k, u)$ and $\hat{p}(z \mid b_k, u)$; cf. Eq. (3). The belief update calculation $F(b_k, z, u)$ similarly benefits from the small number of nonzero components of $b_k$ and the sparsity of the transition probabilities. In particular, the belief estimator $F$ is defined through Bayes' rule, i.e., the $j$th component of $F(b, u, z)$, denoted by $F(b, u, z)(j)$, is given as

$$F(b, u, z)(j) = \frac{p(z \mid j, u) \sum_{i=1}^n b(i) p_{ij}(u)}{\sum_{i=1}^n \sum_{j'=1}^n p(z \mid j', u) p_{ij'}(u) b(i)}. \tag{24}$$

For the same reasons, the number of states $j$ where $F(b_k, u, z)(j) > 0$ is much smaller than $n$. In addition, the numbers of states $i$ and $j'$ to enumerate in Eq. (24) are also much smaller than $n$.

From the above discussion, we can see that the unobservable states $i$ involved in our algorithm form a set $\hat{X}$, which is defined as

$$\hat{X} = \{i \mid i \in \cup_{x \in \mathcal{F}} I_x \text{ or } p_{ji}(u) > 0 \text{ for some } j \in \cup_{x \in \mathcal{F}} I_x \text{ and some } u \in U\}.$$

In fact, one may interpret our algorithm as using the POMDP with the unobservable state space $\hat{X}$ to approximate the original POMDP.

We demonstrate the computational savings from our scheme by using our running example.

**Example 2.3.** *Let us illustrate the cost function approximation $\tilde{J}$ computed via our aggregation method when the set $\tilde{Q}$ and the probabilities $\psi_{q\tilde{q}}$ for all $q \in Q$ and $\tilde{q} \in \tilde{Q}$ are defined according to Eqs. (9) and (10), respectively; cf. Example 2.2. For the computational results to be presented shortly, the problem data and experimental setup are summarized in Appendix A. When applying our feature-based belief aggregation, we use discretization resolution $\rho = 10$ [cf. Eq. (9)], and define as a feature state the index of the site with the biggest undiscovered value; cf. Example 2.1. Fig. 5 shows a comparison between our feature-based aggregation method and standard aggregation (i.e., non-feature-based aggregation, which is the special case of our method where $\mathcal{F} = X$). We note that without feature-based aggregation, the cost function approximation in Eq. (18) becomes computationally impractical when the number of search sites exceeds $N = 4$. By contrast, our feature-based belief aggregation can scale to problems with many more search sites. We also find that feature-based belief aggregation leads to similar approximation error compared to standard aggregation.*

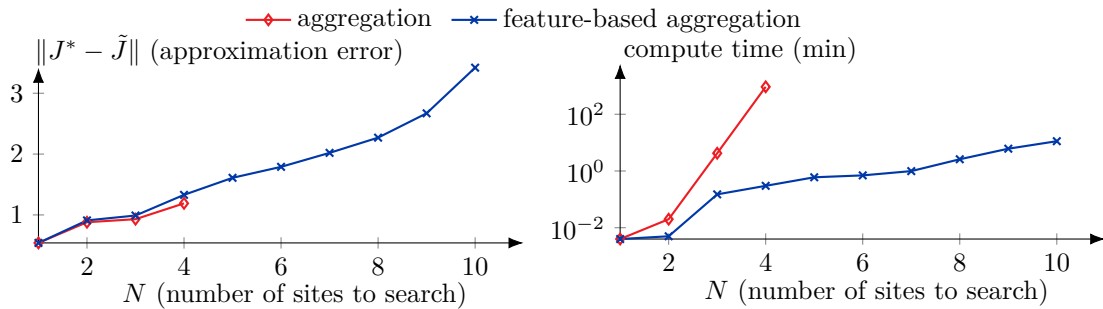

Figure 5: Approximation error and computation time of the cost function approximation $\tilde{J}$ as defined by Eq. (18) for the running example when aggregating the original belief space (red curves) and the feature belief space (blue curves). The horizontal axes indicate the number of search sites $N$, which determines the computational complexity of the problem. For problems with $N > 4$, the computation of the standard aggregation scheme could not be completed within 2 hours, and the corresponding results are not recorded in the figure.

## 3 Analysis of the Approximation Error

In this section, we study the difference between the cost function approximation $\tilde{J}$ of Eq. (18) relative to the optimal cost function $J^*$. We first consider the values $\tilde{J}(b)$ for beliefs in $\tilde{B}$; cf. Eq. (14). Based on the analysis of these values, we quantify the differences between $\tilde{J}$ and the optimal cost $J^*$. Moreover, when the set $\tilde{B}$, the mapping $\Phi$ [cf. Eq. (15)], and the belief aggregation probabilities $\psi$ [cf. Eq. (8)] satisfy certain conditions, we show that $\tilde{J}$ is a lower bound of $J^*$ within a certain error.

### 3.1 Cost Function Approximation for Beliefs in $\tilde{B}$

Starting from some representative feature belief $\tilde{q}$, our method generates deterministically a belief $b \in \tilde{B}$ according to $b = D(\tilde{q})$, where the mapping $D : \tilde{Q} \mapsto B$ is defined in Eq. (13), and $\tilde{B}$ is the image of the

mapping $D$, i.e.,

$$\tilde{B} = \big\{ b \in B \,|\, b = D(\tilde{q}) \text{ for some } \tilde{q} \in \tilde{Q} \big\},$$

cf. Eq. (14). To represent the belief space $B$ sufficiently well by the representative feature belief space $\tilde{Q}$, it is desirable for different representative feature beliefs $\tilde{q}$ and $\tilde{q}'$ to map to different belief states $\tilde{b}$ and $\tilde{b}'$. In other words, the mapping $D$, when viewed as a mapping from $\tilde{Q}$ to $\tilde{B}$, should be one-to-one.[3] Indeed this is the case, thanks to the assumption that the nonempty sets $I_x$ are disjoint, as we show next.

**Proposition 2.** *Let $\tilde{q}$ and $\tilde{q}'$ be two representative feature beliefs that are different, i.e., $\tilde{q}, \tilde{q}' \in \tilde{Q}$ and $\tilde{q} \neq \tilde{q}'$. Then the belief states $b = D(\tilde{q})$ and $b' = D(\tilde{q}')$ are also different.*

*Proof.* For every $x \in \mathcal{F}$, let us select some $i_x \in \{1, 2, \ldots, n\}$ such that $i_x \in I_x$ and $d_{x i_x} > 0$. Since the nonempty sets $I_x$ are disjoint, we have that for every $i_x$,

$$i_x \notin I_{x'}, \qquad \text{for all } x' \neq x \tag{25}$$

Recall that for every feature state $x$, its disaggregation probabilities satisfy $d_{xi} > 0$ only if $i \in I_x$; cf. Eq. (5). Therefore, Eq. (25) implies that for every $i_x$,

$$d_{x' i_x} = 0, \qquad \text{for all } x' \neq x. \tag{26}$$

Since $\tilde{q} \neq \tilde{q}'$, there exists some $\tilde{x} \in \mathcal{F}$ such that $\tilde{q}(\tilde{x}) \neq \tilde{q}'(\tilde{x})$. We will show that $b(i_{\tilde{x}}) \neq b'(i_{\tilde{x}})$, which implies that $b \neq b'$.

Indeed, by definition, we have

$$b(i) = \sum_{x \in \mathcal{F}} \tilde{q}(x) d_{xi}, \quad b'(i) = \sum_{x \in \mathcal{F}} \tilde{q}'(x) d_{xi}, \qquad i = 1, 2, \ldots, n.$$

Let us focus on the component $b(i_{\tilde{x}})$ corresponding to $i_{\tilde{x}} \in I_{\tilde{x}}$. It is given by

$$b(i_{\tilde{x}}) = \sum_{x \in \mathcal{F}} \tilde{q}(x) d_{x i_{\tilde{x}}} = \tilde{q}(\tilde{x}) d_{\tilde{x} i_{\tilde{x}}},$$

where the last equality follows from Eq. (26). Similarly, $b'(i_{\tilde{x}}) = \tilde{q}'(\tilde{x}) d_{\tilde{x} i_{\tilde{x}}}$. Since $\tilde{q}(\tilde{x}) \neq \tilde{q}'(\tilde{x})$, we obtain $b(i_{\tilde{x}}) \neq b'(i_{\tilde{x}})$. $\qquad \square$

Note that without the sets $I_x$ being disjoint, different $\tilde{q}$ and $\tilde{q}'$ may be mapped to the same $b$ via the mapping $D$. Let us provide such an example.

**Example 3.1** (Pathology without $I_x$ being disjoint)**.** *Suppose that representative feature beliefs $\tilde{q}$ and $\tilde{q}'$ satisfy*

$$\tilde{q}(\tilde{x}) \neq \tilde{q}'(\tilde{x}), \quad \tilde{q}(x') \neq \tilde{q}'(x'), \qquad \text{for some } \tilde{x}, \, x', \tag{27}$$

*and*

$$\tilde{q}(x) = \tilde{q}'(x), \qquad \text{for all } x \neq \tilde{x} \text{ and } x \neq x'. \tag{28}$$

*Suppose further that the sets $I_{\tilde{x}}$ and $I_{x'}$ and some state $\tilde{i}$ satisfy*

$$\{\tilde{i}\} = I_{\tilde{x}} = I_{x'}, \qquad \tilde{i} \notin I_x, \quad \text{for all } x \neq \tilde{x} \text{ and } x'' \neq x'. \tag{29}$$

*We claim that $D(\tilde{q}) = D(\tilde{q}')$. Indeed, from Eq. (28), we have*

$$\tilde{q}(\tilde{x}) + \tilde{q}(x') = \tilde{q}'(\tilde{x}) + \tilde{q}'(x'). \tag{30}$$

*Moreover, Eq. (29) implies that*

$$d_{\tilde{x}\tilde{i}} = d_{x'\tilde{i}} = 1, \qquad d_{x\tilde{i}} = 0, \quad \text{for all } x \neq \tilde{x} \text{ and } x'. \tag{31}$$

---

[3]We say a function $f$ that maps the set $\tilde{Q}$ to the set $\tilde{B}$ is one-to-one, if i) for every $\tilde{b}$, there exists some $\tilde{q} \in \tilde{Q}$ such that $f(\tilde{q}) = \tilde{b}$, and ii) $f(\tilde{q}) \neq f(\tilde{q}')$ for every $\tilde{q}, \tilde{q} \in \tilde{Q}$ and $\tilde{q} \neq \tilde{q}'$.

*Consider the component of $D(\tilde{q})$ that corresponds to the state $\tilde{i}$. It is given by*

$$\sum_{x \in \mathcal{F}} \tilde{q}(x)d_{x\tilde{i}} = \tilde{q}(\tilde{x})d_{\tilde{x}\tilde{i}} + \tilde{q}(x')d_{x'\tilde{i}} = \tilde{q}(\tilde{x}) + \tilde{q}(x')$$

$$= \tilde{q}'(\tilde{x}) + \tilde{q}'(x') = \tilde{q}'(\tilde{x})d_{\tilde{x}\tilde{i}} + \tilde{q}'(x')d_{x'\tilde{i}}$$

$$= \sum_{x \in \mathcal{F}} \tilde{q}'(x)d_{x\tilde{i}},$$

*where the first and the second equalities follow from Eq. (31), the third equality is Eq. (30), and the fourth and fifth equalities follow from Eq. (31). The last term $\sum_{x \in \mathcal{F}} \tilde{q}'(x)d_{x\tilde{i}}$ is the component of $D(\tilde{q}')$ that corresponds to the state $\tilde{i}$.*

*As for every other component of $D(\tilde{q})$ that corresponds to some state $i \neq \tilde{i}$, we have*

$$\sum_{x \in \mathcal{F}} \tilde{q}(x)d_{xi} = \sum_{x \in \mathcal{F}, x \neq \tilde{x}, x \neq x'} \tilde{q}(x)d_{xi}$$

$$= \sum_{x \in \mathcal{F}, x \neq \tilde{x}, x \neq x'} \tilde{q}'(x)d_{xi} = \sum_{x \in \mathcal{F}} \tilde{q}'(x)d_{xi},$$

*where the first and the last equality follow from the condition $\{\tilde{i}\} = I_{\tilde{x}} = I_{x'}$ and Eq. (28). Clearly, the last term is the component of $D(\tilde{q}')$ that corresponds to $i$. Therefore, we have $D(\tilde{q}) = D(\tilde{q}')$.*

The preceding proposition states that through the disaggregation process, each representative feature belief $\tilde{q}$ maps to a unique belief state $b$ that belongs to the set $\tilde{B}$. In other words, the mapping $D$, when viewed as a mapping from $\tilde{Q}$ to $\tilde{B}$, is one-to-one. As our next result, we show that the aggregation process defines the inverse of such a mapping.

**Proposition 3.** *Let $b$ be a belief state belonging to the set $\tilde{B}$ and $\tilde{q}$ be a representative belief. Then $b = D(\tilde{q})$ if and only if $\tilde{q} = \Phi(b)$, i.e.,*

$$b(i) = \sum_{x \in \mathcal{F}} \tilde{q}(x)d_{xi}, \qquad i = 1, 2, \dots, n,$$

*if and only if*

$$\tilde{q}(y) = \sum_{j=1}^{n} b(j)\phi_{jy}, \qquad \text{for all } y \in \mathcal{F}.$$

*Proof.* We first prove the only if part. Let us denote by $q$ the vector $\Phi(b)$. In what follows, we will show that $q = \tilde{q}$. By applying the relation $b = D(\tilde{q})$, we have for every $y \in \mathcal{F}$,

$$q(y) = \sum_{j=1}^{n} b(j)\phi_{jy} = \sum_{j=1}^{n} \sum_{x \in \mathcal{F}} \tilde{q}(x)d_{xj}\phi_{jy}$$

$$= \sum_{x \in \mathcal{F}} \tilde{q}(x) \sum_{j=1}^{n} d_{xj}\phi_{jy} = \sum_{x \in \mathcal{F}} \tilde{q}(x) \sum_{j \in I_x} d_{xj}\phi_{jy}, \tag{32}$$

where the last equality holds because $d_{xj} = 0$ for all $j \notin I_x$; cf. Eq. (5).

Let us now focus on the sum $\sum_{j \in I_x} d_{xj}\phi_{jy}$. We claim that

$$\sum_{j \in I_x} d_{xj}\phi_{jy} = \begin{cases} 1 & \text{if } x = y, \\ 0 & \text{otherwise.} \end{cases} \tag{33}$$

Suppose $x = y$. We have $\phi_{jy} = 1$ for all $j \in I_x = I_y$ according to Eq. (6). Since $\sum_{j \in I_x} d_{xj} = \sum_{j=1}^{n} d_{xj} = 1$, we obtain the desired equality. Suppose $x \neq y$. Using the fact that $I_x$ and $I_y$ are disjoint and that $\phi_{jx} = 1$ for $j \in I_x$, we have $\phi_{jy} = 0$ for $j \in I_x$. Therefore, $\sum_{j \in I_x} d_{xj}\phi_{jy} = 0$. Combining Eq. (32) with Eq. (33) leads to the equality $q(y) = \tilde{q}(y)$. Since $y$ is arbitrary, we obtain $q = \tilde{q}$.

Next, we show the if part. Suppose $\tilde{q} = \Phi(b)$. Since $b \in \tilde{B}$, there exists some $\tilde{q}'$ such that $b = D(\tilde{q}')$. By the only if of the statement, the equality $b = D(\tilde{q}')$ implies $\tilde{q}' = \Phi(b)$. However, $\tilde{q} = \Phi(b)$ by assumption. Therefore, $\tilde{q}' = \tilde{q}$, or equivalently, $\tilde{q} = \Phi(b)$. $\qquad\square$

Having investigated the properties of the mappings $D$ and $\Phi$, we are ready to characterize the cost function approximation $\tilde{J}$ for beliefs in $\tilde{B}$.

**Proposition 4.** *Let $b$ be a belief state belonging to the set $\tilde{B}$ such that $b = D(\tilde{q})$ for some $\tilde{q} \in \tilde{Q}$. We have*

$$\psi_{\Phi(b)\tilde{p}} = 1, \qquad \psi_{\Phi(b)\tilde{p}'} = 0, \quad \text{for all } \tilde{p}' \neq \tilde{p}. \tag{34}$$

*Moreover,*

$$\tilde{J}(b) = r_{\tilde{q}}^*.$$

*Proof.* By Prop. 3, $b = D(\tilde{q})$ implies $\tilde{q} = \Phi(b)$. In view of the definitions of belief aggregation probability $\psi$ [cf. Eq. (8)], we obtain Eq. (34). Moreover,

$$\tilde{J}(b) = \sum_{\tilde{q}' \in \tilde{Q}} \psi_{\Phi(b)\tilde{q}'} r_{\tilde{q}'}^* = \sum_{\tilde{q}' \in \tilde{Q}} \psi_{\tilde{q}\tilde{q}'} r_{\tilde{q}'}^* = r_{\tilde{q}}^*,$$

where the last equality follows from Eq. (34). $\qquad\square$

In summary, for every $b$ such that $b = D(\tilde{q})$ for some $\tilde{q}$, its corresponding cost function approximation value $\tilde{J}(b)$ depends solely on the component of $r^*$ that corresponds to the representative feature belief $\tilde{q}$.

## 3.2 Error Bound of The Cost Function Approximation $\tilde{J}$

We will now quantify the difference between the cost function approximation $\tilde{J}$ and the optimal cost function $J^*$; cf. Eqs. (18) and (4). To this end, we define a *footprint set* of each representative feature belief $\tilde{q} \in \tilde{Q}$ as

$$S_{\tilde{q}} = \{b \in B \mid \psi_{\Phi(b)\tilde{q}} > 0\}. \tag{35}$$

In other words, for every belief state $b \in S_{\tilde{q}}$, the cost function approximation $\tilde{J}(b)$ depends on $r_{\tilde{q}}^*$ with weight $\psi_{\Phi(b)\tilde{q}}$; cf. Eq. (8). Similarly, for every belief state $b \in B$, we associate a *representative set* $R_b$ defined as

$$R_b = \{\tilde{q} \in \tilde{Q} \mid \psi_{\Phi(b)\tilde{q}} > 0\}. \tag{36}$$

In view of Props. 3 and 4, we have

$$D(\tilde{q}) \in S_{\tilde{q}}, \qquad \text{for all } \tilde{q} \in \tilde{Q}. \tag{37}$$

We now characterize the approximation error of $\tilde{J}$. Our analysis relies critically on the condition (37).

**Proposition 5.** *The cost function approximation $\tilde{J}$ defined in Eq. (18) and the optimal cost function $J^*$ satisfy*

$$|\tilde{J}(b) - J^*(b)| \leq \frac{\epsilon}{1 - \alpha}, \qquad \text{for all } b \in B, \tag{38}$$

*where $\epsilon$ is the scalar defined by*

$$\epsilon = \max_{\tilde{q} \in \tilde{Q}} \sup_{b, b' \in S_{\tilde{q}}} |J^*(b) - J^*(b')|. \tag{39}$$

*Proof.* First, we show that $\epsilon$ is finite. Since the number of states and controls is finite and the cost function $g$ is bounded, it follows that the function $\hat{g}$ in Eq. (3) is bounded. Thus, from the theory of discounted cost DP problems, it follows that $J^*$ is bounded; see (Bertsekas, 2012, Section 1.2). As a result, for every $\tilde{q}$, $\sup_{b, b' \in S_{\tilde{q}}} |J^*(b) - J^*(b')|$ is finite, and the finiteness of $\epsilon$ follows from the finiteness of the set $\tilde{Q}$.

Next, we consider the operator $H$ defined by Eq. (16) and the $m$-dimensional vector $\bar{r}$ with components defined by

$$\bar{r}_{\tilde{q}} = \inf_{b \in S_{\tilde{q}}} J^*(b) + \frac{\epsilon}{1-\alpha}, \qquad \tilde{q} \in \tilde{Q}. \tag{40}$$

These components are finite, and therefore, we have $\bar{r} \in \Re^m$. In view of the definitions of $S_{\tilde{q}}$, $R_b$, and $\bar{r}_{\tilde{q}}$ given in Eqs. (35), (36), and (40) we have

$$\bar{r}_{\tilde{q}} \le J^*(b) + \frac{\epsilon}{1-\alpha}, \qquad \text{for all } \tilde{q} \in R_b,\, b \in B. \tag{41}$$

For every $\tilde{q} \in \tilde{Q}$, we have

$$
\begin{aligned}
&(H\bar{r})(\tilde{q}) \\
&= \min_{u \in U} \left[ \hat{g}\big(D(\tilde{q}), u\big) + \alpha \sum_{z \in Z} \hat{p}\big(z \mid D(\tilde{q}), u\big) \sum_{\tilde{q}' \in \tilde{Q}} \psi_{G(\tilde{q}, u, z)\tilde{q}'} \bar{r}_{\tilde{q}'} \right] \\
&= \min_{u \in U} \left[ \hat{g}\big(D(\tilde{q}), u\big) + \alpha \sum_{z \in Z} \hat{p}\big(z \mid D(\tilde{q}), u\big) \sum_{\tilde{q}' \in \tilde{Q}} \psi_{\Phi\big(F(D(\tilde{q}), u, z)\big)\tilde{q}'} \bar{r}_{\tilde{q}'} \right] \\
&= \min_{u \in U} \left[ \hat{g}\big(D(\tilde{q}), u\big) + \alpha \sum_{z \in Z} \hat{p}\big(z \mid D(\tilde{q}), u\big) \sum_{\tilde{q}' \in R_{F(D(\tilde{q}), u, z)}} \psi_{\Phi\big(F(D(\tilde{q}), u, z)\big)\tilde{q}'} \bar{r}_{\tilde{q}'} \right] \\
&\le \min_{u \in U} \left[ \hat{g}\big(D(\tilde{q}), u\big) + \alpha \sum_{z \in Z} \hat{p}\big(z \mid D(\tilde{q}), u\big) J^*\big(F(D(\tilde{q}), u, z)\big) \right] + \frac{\alpha\epsilon}{1-\alpha} \\
&= J^*\big(D(\tilde{q})\big) + \frac{\alpha\epsilon}{1-\alpha} \\
&\le \inf_{b \in S_{\tilde{q}}} J^*(b) + \epsilon + \frac{\alpha\epsilon}{1-\alpha} \\
&= \bar{r}_{\tilde{q}},
\end{aligned}
$$

where the second equality is due to the definition of $G$; cf. Eq. (17). The third equality is due to the definition of $R_{F\big(D(\tilde{q}), u, z\big)}$; cf. Eq. (36). The first inequality follows from Eq. (41) with $F\big(D(\tilde{q}), u, z\big)$ in place of $b$ and the fact that $\sum_{z \in Z} \hat{p}\big(z \mid D(\tilde{q}), u\big) = 1$. The fourth equality holds because $J^*$ satisfies Bellman's equation of the original problem; cf. Eq. (4). The last inequality follows from the definition of $\epsilon$ and the fact that $D(\tilde{q}) \in S_{\tilde{q}}$, as stated in Eq. (37); cf. Eq. (39). From this inequality, we obtain $H\bar{r} \le \bar{r}$.

Due to the monotonicity of $H$ [cf. Prop. 1], we have that the sequence $\{r^k\}$ with $r^0 = \bar{r}$ and $r^{k+1} = Hr^k$ is monotonically decreasing. Moreover, from the contractivity of $H$ [cf. Prop. 1], we have $\lim_{k \to \infty} r^k = r^*$. Combining these properties with Eq. (41), we obtain

$$r_{\tilde{q}}^* \le \bar{r}_{\tilde{q}} \le J^*(b) + \frac{\epsilon}{1-\alpha}, \qquad \text{for all } \tilde{q} \in R_b,\, b \in B.$$

As a result, we have

$$\tilde{J}(b) = \sum_{\tilde{q} \in \tilde{Q}} \psi_{\Phi(b)\tilde{q}} r_{\tilde{q}}^* = \sum_{\tilde{q} \in R_b} \psi_{\Phi(b)\tilde{q}} r_{\tilde{q}}^* \le J^*(b) + \frac{\epsilon}{1-\alpha}.$$

The converse inequality can be shown by considering the $m$-dimensional vector $\underline{r}$ with components

$$\underline{r}_{\tilde{q}} = \sup_{b \in S_{\tilde{q}}} J^*(b) - \frac{\epsilon}{1-\alpha}, \qquad \tilde{q} \in \tilde{Q}.$$

The proof is thus complete. $\qquad \square$

The scalar $\epsilon$ of Eq. (39) is the maximum variation of the optimal cost function $J^*$ within the footprint sets $\{S_{\tilde{q}} \mid \tilde{q} \in \tilde{Q}\}$; see Fig. 6. Thus, the meaning of the preceding proposition is that if $J^*$ varies by at most $\epsilon$

within each footprint set, then the interpolation formula in Eq. (18) yields a cost function approximation that is within $\frac{\epsilon}{1-\alpha}$ of the optimal.

Consistent with the aggregation literature, Tsitsiklis & van Roy (1996), (Bertsekas, 2012, Section 6.5), and Bertsekas (2018), the bound in Prop. 5 is not intended as a computable numerical quantity but as an analytical characterization of how the approximation error depends on the aggregation structure, specifically, on the variation of $J^*$ within each footprint set. This interpretation clarifies the source of approximation error and provides guidance for effective choices of design parameters in our method. In particular, we seek a feature space $\mathcal{F}$, disaggregation probabilities $d_{xj}$, and aggregation probabilities $\phi_{jx}$ so that the optimal cost $J^*$ is approximately constant over the sets $S_{\tilde{q}}$. The bound also relies on mild, natural yet necessary conditions [(a)–(d) in Section 2.1] that ensure the model is well posed; omitting these can lead to pathological cases such as those illustrated in Example 3.1.

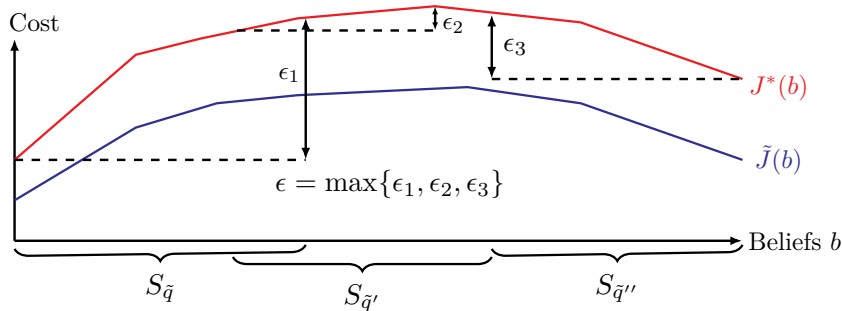

Figure 6: Schematic illustration of the scalar $\epsilon$ of Eq. (39) in Prop. 5; the illustration is based on an approximation with representative feature beliefs $\tilde{q}, \tilde{q}'$, and $\tilde{q}''$, with footprint sets $S_{\tilde{q}}, S_{\tilde{q}'}$, and $S_{\tilde{q}''}$, respectively; cf. Eq. (35).

Note that Prop. 5 places no restrictions on the aggregation probabilities $\phi_{jy}$ nor the belief aggregation probabilities $\psi_{q\tilde{q}}$, except those specified by Eqs. (6) and (8). Consequently, the error bound in Prop. 5 goes beyond the classical bound by Tsitsiklis and van Roy (Tsitsiklis & van Roy, 1996), where it is assumed that the collection of the footprint sets forms a partition of the (observable) state space. For our aggregation method, the conditions stated in Tsitsiklis & van Roy (1996) imply that for every $b \in B$, there exists exactly one $\tilde{q} \in \tilde{Q}$ such that $\psi_{\Phi(b)\tilde{q}} = 1$ and $\psi_{\Phi(b)\tilde{q}'} = 0$ for all $\tilde{q}' \neq \tilde{q}$. A sufficient condition for this to hold is that for every $q \in Q$, there exits exactly one $\tilde{q} \in \tilde{Q}$ such that $\psi_{q\tilde{q}} = 1$ and $\psi_{q\tilde{q}'} = 0$ for all $\tilde{q}' \neq \tilde{q}$. In contrast, the bound in Eq. (38) allows states and feature beliefs to aggregate to multiple feature states and representative feature beliefs, respectively. This implies that the footprint sets $S_{\tilde{q}}$ and $S_{\tilde{q}'}$ that correspond to different representative feature beliefs may have nonempty intersections, as indicated in Fig. 6. Moreover, our proof does not rely on properties specific to POMDP, allowing it to be adopted for similar aggregation methods for MDP involving only observable states. A version of this proof, tailored specifically to MDPs and featuring a generalized form of condition (37), is presented in Anonymous (2025b).

While the bound in Prop. 5 provides qualitative insight, it is conservative and smaller approximation errors can be expected in practice. We illustrate this by using our running example.

**Example 3.2.** *Let us consider the treasure hunting problem involving a single site, i.e., $N = 1$, so that we can plot the values of $J^*$ and $\tilde{J}$. We apply our aggregation scheme with the features described in Example 2.1. Since there is only one site involved, we have $\mathcal{F} = X$. The representative feature beliefs $\tilde{Q}$ and the belief aggregation probabilities are defined as in Example 2.3; cf. Eqs. (9) and (10) in Example 2.2. The problem data are given in Appendix A.*

*We show in Fig. 7 a comparison between the cost function approximation $\tilde{J}$ and the optimal cost function $J^*$ [cf. Eq. (4)] for varying discretization resolutions $\rho$ [cf. Eq. (9)]. We note that the difference between the cost function approximation $\tilde{J}$ and the optimal cost $J^*$ is reduced as the discretization resolution $\rho$ is increased, as expected from the error bound of Prop. 5.*

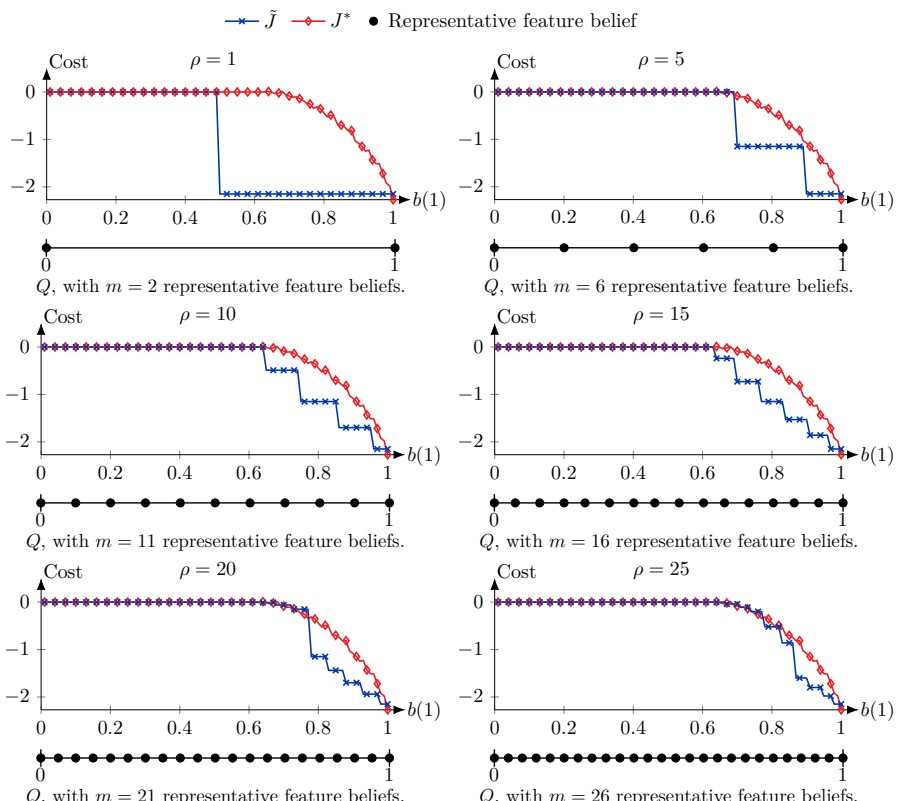

Figure 7: Comparison between the optimal cost function $J^*$ [cf. Eq. (4)] and the cost function approximation $\tilde{J}$ [cf. Eq. (18)] for the POMDP for the treasure hunting problem when the number of sites $N$ is 1, and the belief aggregation probabilities $\psi_{q\tilde{q}}$ are defined based on Eq. (10). The horizontal axes indicate the belief that the site has a treasure, and the vertical axes show the corresponding expected cost. Each plot relates to a different discretization resolution $\rho$ [cf. Eq. (9)], which leads to a different number $m$ of representative feature beliefs, as indicated below the horizontal axes.

*We show in Fig. 8 a comparison between the theoretical bound and the actual approximation error of the cost function approximation $\tilde{J}$. We observe that the bound is not tight (as remarked earlier) but gets closer to the actual error when the discretization resolution increases. Similar patterns can be observed for larger values of $N$.*

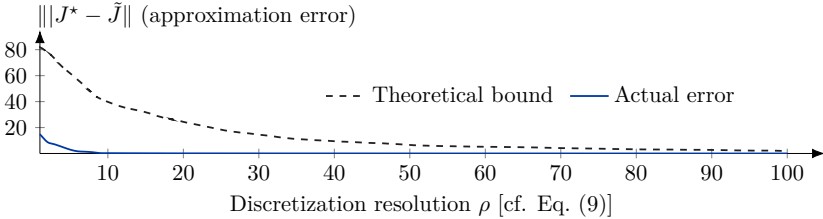

Figure 8: Comparison between the theoretical error bound [cf. Prop. 5] and the actual error of the cost function approximation $\tilde{J}$ [cf. (18)] when applied to the treasure hunting example with a single search site, i.e., $N = 1$.

### 3.3 The Cost Function Approximation as a Lower Bound

In the preceding section, we quantified the difference between the cost function approximation $\tilde{J}$ [cf. Eq. (18)] and the optimal cost function $J^*$. Within the context of POMDP, it is well-known [cf. Yu & Bertsekas (2004) and (Bertsekas, 2012, Example 6.5.5)] that a cost function approximation constructed with a suitably defined aggregation scheme can serve as a lower bound of the optimal cost function $J^*$. In what follows, we delineate conditions under which the cost function approximation $\tilde{J}$ computed via our approach can serve as a lower bound of $J^*$.

In particular, we require that the disaggregation, aggregation, and belief aggregation probabilities satisfy

$$b = \sum_{b' \in \tilde{B}} \psi_{\Phi(b)\Phi(b')} b', \qquad \text{for all } b \in B. \tag{42}$$

In other words, the convex hull of $\tilde{B}$ [cf. Eq. (14)] equals $B$, and the convex combination coefficient associated with $b \in B$ and $b' \in \tilde{B}$ is the belief aggregation probability $\psi_{\Phi(b)\Phi(b')}$. Under this condition, we have the following result.

**Proposition 6.** *Let the disaggregation, aggregation, and belief aggregation probabilities satisfy Eq. (42). The cost function approximation $\tilde{J}$ defined in Eq. (18) and the optimal cost function $J^*$ satisfy*

$$0 \le J^*(b) - \tilde{J}(b) \le \frac{\epsilon}{1 - \alpha}, \qquad \text{for all } b \in B, \tag{43}$$

*where $\epsilon$ is given by Eq. (39). In addition, if $\tilde{Q}$ contains $n$ representative feature beliefs, $\tilde{J}$ is linear.*

*Proof.* In what follows, we prove that $\tilde{J}(b) \le J^*(b)$ for all $b \in B$. The other side of Eq. (43) follows directly from Prop. 5. From Prop. 3, we have that for every $b \in \tilde{B}$, $b = D(\tilde{q})$ if and only if $\tilde{q} = \Phi(b)$. As a result, the relation given in Eq. (42) can be stated equivalently as

$$b = \sum_{\tilde{q} \in \tilde{Q}} \psi_{\Phi(b)\tilde{q}} D(\tilde{q}), \qquad \text{for all } b \in B. \tag{44}$$

It is well-known that the optimal cost function $J^*$ is concave in $b$; see, e.g., (Sondik, 1978, Thm. 2), (Krishnamurthy, 2016, Thms. 7.6.1-7.6.2), or (Bertsekas, 2012, Section 6.5.1). The concavity of $J^*$ and Eq. (44) imply that

$$J^*(b) = J^* \left( \sum_{\tilde{q} \in \tilde{Q}} \psi_{\Phi(b)\tilde{q}} D(\tilde{q}) \right) \ge \sum_{\tilde{q} \in \tilde{Q}} \psi_{\Phi(b)\tilde{q}} J^* \big( D(\tilde{q}) \big), \qquad \text{for all } b \in B. \tag{45}$$

Since $J^*$ also satisfies Bellman's equation of the POMDP [cf. Eq. (4)], we obtain

$$
\begin{aligned}
J^*(b) &= \min_{u \in U} \left[ \hat{g}(b, u) + \alpha \sum_{z \in Z} \hat{p}(z \mid b, u) J^* \big( F(b, u, z) \big) \right] \\
&= \min_{u \in U} \left[ \hat{g}(b, u) + \alpha \sum_{z \in Z} \hat{p}(z \mid b, u) J^* \left( \sum_{\tilde{q} \in \tilde{Q}} \psi_{\Phi(F(b,u,z))\tilde{q}} D(\tilde{q}) \right) \right] \\
&\ge \min_{u \in U} \left[ \hat{g}(b, u) + \alpha \sum_{z \in Z} \hat{p}(z \mid b, u) \sum_{\tilde{q} \in \tilde{Q}} \psi_{\Phi(F(b,u,z))\tilde{q}} J^* \big( D(\tilde{q}) \big) \right],
\end{aligned}
\tag{46}
$$

where the second equality follows from Eq. (44) with $F(b, u, z)$ in place of $b$, and the inequality follows from Eq. (45) with $F(b, u, z)$ in place of $b$. In particular, for every $b \in \tilde{B}$ so that $b = D(\tilde{q})$ for some $\tilde{q} \in \tilde{Q}$, Eq. (46) becomes

$$J^* \big( D(\tilde{q}) \big) \ge \min_{u \in U} \left[ \hat{g} \big( D(\tilde{q}), u \big) + \alpha \sum_{z \in Z} \hat{p} \big( z \mid D(\tilde{q}), u \big) \sum_{\tilde{q}' \in \tilde{Q}} \psi_{G(\tilde{q}, u, z)\tilde{q}'} J^* \big( D(\tilde{q}') \big) \right]. \tag{47}$$

From the definition of $\tilde{B}$, we have that Eq. (47) holds for all $\tilde{q} \in \tilde{Q}$.

Let us consider a vector $\hat{r} \in \Re^m$ with components defined by

$$\hat{r}_{\tilde{q}} = J^*\big(D(\tilde{q})\big), \qquad \tilde{q} \in \tilde{Q}.$$

By definition of $H$ [cf. Eq. (16)], Eq. (47) can be written equivalently as

$$H\hat{r} \geq \hat{r}. \tag{48}$$

Using arguments similar to those in the proof for Prop. 5, Eq. (48) implies that $\hat{r} \geq r^*$, where $r^*$ is the fixed point of $H$. Then for every $b \in B$, we have

$$\tilde{J}(b) = \sum_{\tilde{q} \in \tilde{Q}} \psi_{\Phi(b)\tilde{q}} r^*_{\tilde{q}} \leq \sum_{\tilde{q} \in \tilde{Q}} \psi_{\Phi(b)\tilde{q}} \hat{r}_{\tilde{q}} = \sum_{\tilde{q} \in \tilde{Q}} \psi_{\Phi(b)\tilde{q}} J^*\big(D(\tilde{q})\big).$$

Combining this inequality with Eq. (45) yields the desired inequality.

Suppose that $\tilde{Q}$ contains $n$ representative feature beliefs. Since the set $\tilde{B}$ spans $B$ [cf. Eq. (42)], and $B$ is the belief space over $n$ states, we have that $\tilde{B}$ is linearly independent; cf. Eq. (14). Consider three belief states $b_1$, $b_2$, and $b_3$, where $b_3 = \gamma b_1 + (1 - \gamma)b_2$ and $\gamma \in [0, 1]$. Due to the linear independence of $\tilde{B}$, the aggregation probability $\psi_{\Phi(b_3)\Phi(b')}$ of $b_3$ satisfies

$$\psi_{\Phi(b_3)\Phi(b')} = \gamma \psi_{\Phi(b_1)\Phi(b')} + (1 - \gamma)\psi_{\Phi(b_2)\Phi(b')}, \qquad b' \in \tilde{B}.$$

Therefore, we have $\tilde{J}(b_3) = \gamma \tilde{J}(b_1) + (1 - \gamma)\tilde{J}(b_2)$. □

**Example 3.3.** *Let us illustrate Prop. 6 through the treasure hunting problem in Example 1.1 with a single search site. We construct the representative feature belief space using the method described in Example 3.2, employing varying discretization resolutions. The belief aggregation probabilities $\psi_{q\tilde{q}}$ are computed to satisfy the condition given in Eq. (42). Additional computational details are provided in Appendix A.*

*We show in Fig. 9 a comparison between the cost function approximation $\tilde{J}$ and the optimal cost function $J^*$ [cf. Eq. (4)] for varying discretization resolutions $\rho$ [cf. Eq. (9)]. We note that the cost function approximations $\tilde{J}$ are lower bounds to the optimal cost, i.e., $\tilde{J} \leq J^*$, as asserted in Prop. 6. In contrast, the cost function approximations computed in Example 3.2 may not be lower bounds of $J^*$, as shown in Fig. 7. This is because the belief aggregation probabilities used in Example 3.2 do not satisfy the conditions in Eq. (44). Moreover, when $\rho = 1$, the function $\tilde{J}$ computed here becomes a linear function of $b$.*

## 4 Computational Experiments

We evaluate our feature-based aggregation method on a well-known problem in the POMDP literature, namely the Rocksample (RS) problem Smith & Simmons (2004) with varying sizes of the unobservable state space. We also evaluate it on a new problem: Cyber Autonomy Gym for Experimentation 2 (CAGE-2) CAGE (2022), which involves a much larger unobservable state space and observation space. The sizes of the tested problems are summarized in Table 1.

We chose these two problems for the following reasons.[4] The RS problem serves as the de facto standard benchmark in the POMDP literature and allows us to compare our method with other methods proposed in prior research. In contrast, CAGE-2 is a more recent and substantially larger-scale POMDP that captures the complexity of realistic decision-making environments. Specifically, CAGE-2 is derived from a use case involving the operation and defense of a networked system against cyberattacks, which is an active topic of investigation in the cybersecurity research community. As a consequence, CAGE-2 enables us to assess the applicability of our method to a problem with high practical relevance.

---

[4]We also evaluated the performance of our method on the classical Tiger problem studied in Kaelbling et al. (1998). Our method obtains the optimal cost function when using a discretization resolution of $\rho = 100$. Comparable performance (in terms of both cost and computation time) is achieved by PBVI, SARSOP, POMCP, HSVI, and AdaOps. Other methods either yield suboptimal solutions or require substantially longer computation times to reach optimality.

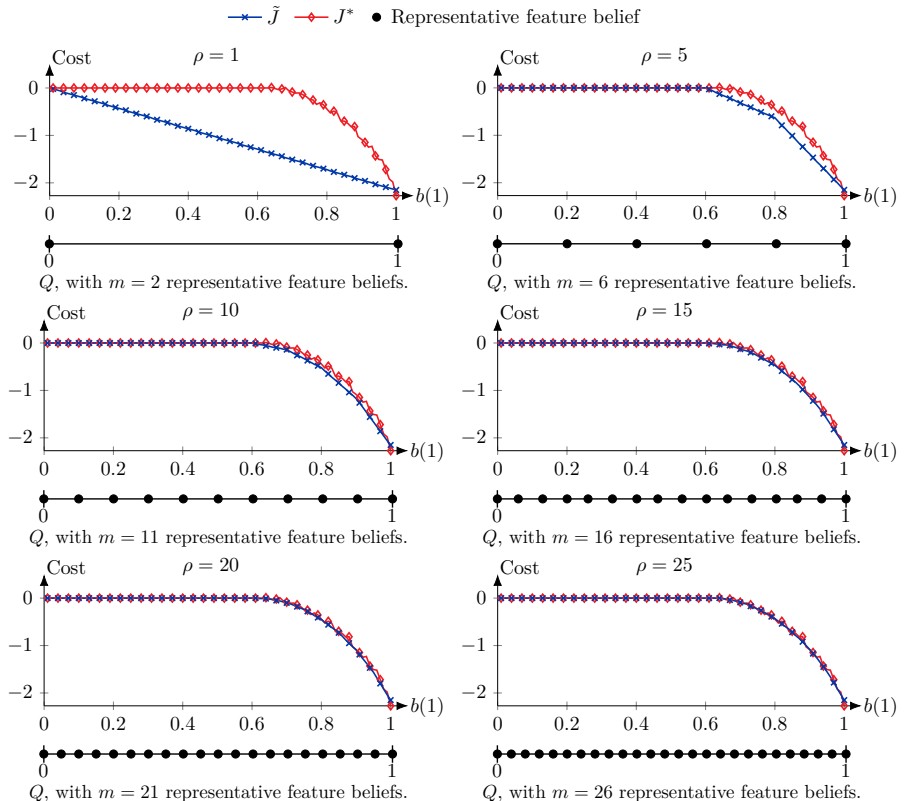

Figure 9: Comparison between the optimal cost function $J^*$ [cf. Eq. (4)] and the cost function approximation $\tilde{J}$ [Eq. (18)] for the POMDP of Example 3.3 when the belief aggregation probabilities $\psi_{q\tilde{q}}$ satisfy Eq. (42). The horizontal axes indicate the belief that the site has a treasure, and the vertical axes show the corresponding expected cost. Each plot relates to a different discretization resolution $\rho$ [cf. Eq. (9)], which leads to a different number $m$ of representative feature beliefs, as indicated below the horizontal axes.

| POMDP | States $n$ | Observations $|Z|$ | Controls $|U|$ |
|---|---|---|---|
| RS (4,4) Smith & Simmons (2004) | 257 | 2 | 9 |
| RS (5,5) Smith & Simmons (2004) | 801 | 2 | 10 |
| RS (5,7) Smith & Simmons (2004) | 3201 | 2 | 12 |
| RS (7,8) Smith & Simmons (2004) | 12545 | 2 | 13 |
| RS (10,10) Smith & Simmons (2004) | 102401 | 2 | 15 |
| CAGE-2 CAGE (2022) | $\geq 10^{47}$ | $\geq 10^{25}$ | 145 |

Table 1: POMDP used for the experimental evaluation, where $|Z|$ and $|U|$ represent the numbers of different observations and controls, respectively.

For the problems considered here, it is not tractable to compute their optimal cost functions $J^*$, making direct comparison between the optimal cost functions and their approximations infeasible. Instead, we test the quality of the cost function approximations $\tilde{J}$ obtained via our method, as well as some value-based methods discussed in Section 1, by applying the policy $\tilde{\mu}$ defined by the following minimization:

$$\tilde{\mu}(b) \in \arg\min_{u \in U} \left[ \hat{g}(b,u) + \alpha \sum_{z \in Z} \hat{p}(z \mid b, u) \tilde{J}(F(b,u,z)) \right];$$

cf. Eq. (19). We also apply the policies computed by heuristic search Silver & Veness (2010); Somani et al. (2013); Sunberg & Kochenderfer (2018); Wu et al. (2021), and the policy-based methods Schulman et al.

(2017); Cobbe et al. (2021). A summary of the tested methods is given in Table 2. Additional problem data and features used in our scheme are given in Appendix A. The code is available at Anonymous (2025a).

| Method | Value-based | Heuristic search | Policy-based |
|---|:---:|:---:|:---:|
| Lovejoy (1991) | ✓ | | |
| QMDP Littman et al. (1995) | ✓ | | |
| HSVI Smith & Simmons (2004) | ✓ | | |
| Roy et al. (2005) | ✓ | | |
| PBVI Pineau et al. (2006) | ✓ | | |
| SARSOP Kurniawati et al. (2008) | ✓ | | |
| POMCP Silver & Veness (2010) | | ✓ | |
| R-DESPOT Somani et al. (2013) | | ✓ | |
| POMCPOW Sunberg & Kochenderfer (2018) | | ✓ | |
| AdaOPS Wu et al. (2021) | | ✓ | |
| PPO Schulman et al. (2017) | | | ✓ |
| PPG Cobbe et al. (2021) | | | ✓ |

Table 2: Methods used for the experimental evaluation.

## 4.1 Evaluation Results On the RS Problems

We present the experimental results for the RS problems according to the categorization in Table 2. Fig. 10 compares our method with value-based approximation approaches on RS problems of varying sizes. Our method requires substantially less off-line computation while achieving competitive performance on small RS instances and outperforming all baselines on large-scale instances. In Fig 11, we compare our method against heuristic search methods. We find that our method performs competitively with leading search methods, despite not relying on extensive on-line computation per stage. Lastly, in Fig. 12, we compare our method with the policy-based approximation methods. We find that our method consistently achieves lower costs than the policy-based methods while being significantly more computationally efficient. In our experiments, both PPO and PPG converge to the same suboptimal policy in the RS problem. This is likely because both methods are policy-gradient algorithms that optimize parameterized policies through gradient-based updates, which can cause them to become trapped in similar local optima. Since we observe that PPO and PPG converge to different policies in the CAGE-2 problem (as detailed in the next subsection), another explanation may be that the RS problem admits only a few reasonable policies, which limits the diversity of possible policies that PPO and PPG can converge to.

## 4.2 Evaluation Results On the CAGE-2 Problem

Due to the large scale of the CAGE-2 POMDP, we were only able to compare our method against three other methods: POMCP Silver & Veness (2010), PPO Schulman et al. (2017), and PPG Cobbe et al. (2021). The other methods are either infeasible for CAGE-2 due to its scale or lack publicly available implementations. The evaluation results are summarized in Fig. 13. The on-line computational times required for our method and POMCP are 0.95 and 30 seconds, respectively. We find that our method obtains the lowest cost among the methods considered, while requiring significantly less off-line computation compared with PPO and PPG. In particular, it attains top performance with only 8.5 minutes of off-line computation without requiring extensive on-line computation per stage, as POMCP does. We attribute this strong performance in part to our choice of features, which are tailored to the structure of the problem; see Appendix A for details. A focused investigation of our method for this problem, where we also integrate other approximation in value space schemes, can be found in Anonymous (2025c).

## 4.3 Discussion of Experimental Results

When compared with other value-based methods, our aggregation method scales well and provides good performance. In particular, HSVI and SARSOP achieve similar or lower costs in small-scale instances of RS, as shown in the first three columns of Fig. 10. However, they become computationally prohibitive for large-scale instances, as well as the CAGE-2 POMDP. The remaining value-based approaches, except for

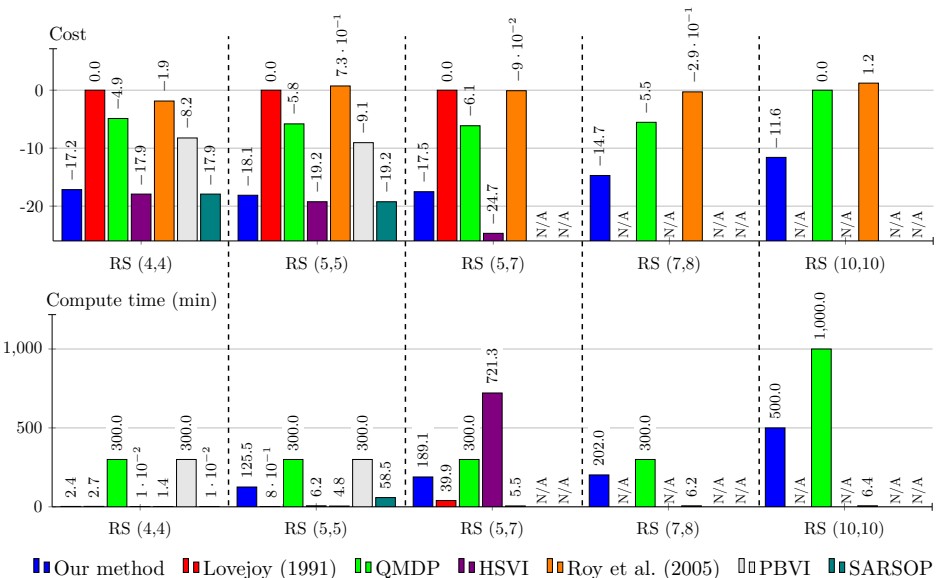

Figure 10: Comparison between our method and value-based approximation methods for solving the RS problems of varying sizes. "N/A" and a missing bar mean that a result could not be obtained after 1,000 minutes of computation.

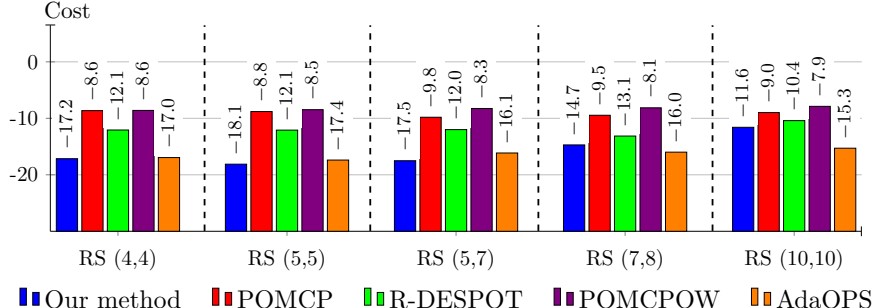

Figure 11: Comparison between our method and heuristic search methods for solving the RS problems with varying sizes. The search methods were given 1 second of search time per stage.

the one proposed in Roy et al. (2005), have higher cost function values in all RS problems while requiring more off-line computation when compared against our method. The method proposed in Roy et al. (2005) requires much less off-line computation than our approach in RS problems. However, this is at the expense of much larger cost values. None of the value-based methods other than our aggregation approach can be implemented in CAGE-2 due to off-line computational challenges, even including that in Roy et al. (2005).

When compared with heuristic search methods in RS problems, POMCP, R-DESPOT, and POMCPPOW have inferior performances to our method; see Fig. 11. AdaOPS has slightly lower cost function values than our method, as shown in Fig. 11. However, it relies on extensive online computations related to the belief estimator. As a result, AdaOPS becomes computationally infeasible for CAGE-2. In fact, the only heuristic search approach we managed to compare against in CAGE-2 is POMCP. However, POMCP yields higher cost function values than our method.

As for policy-based methods, i.e., PPO and PPG, we can see in Figs. 12 and 13 that they have inferior performance while requiring more computational time in RS problems and CAGE-2, when compared with our method.

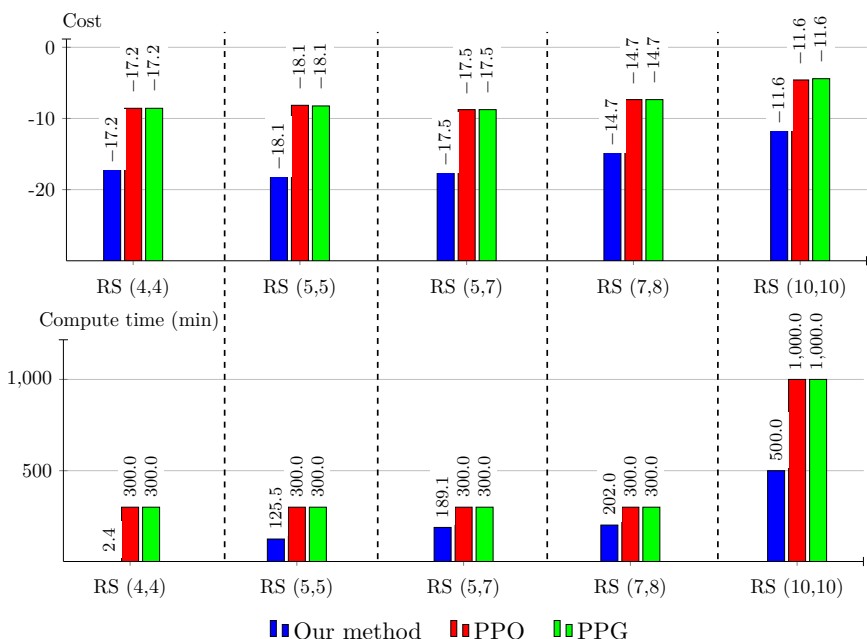

Figure 12: Comparison between our method and policy-based approximation methods for solving the RS problems of varying sizes. The compute time for PPO and PPG indicates the time it took to converge, with a maximum compute time of 1000 minutes.

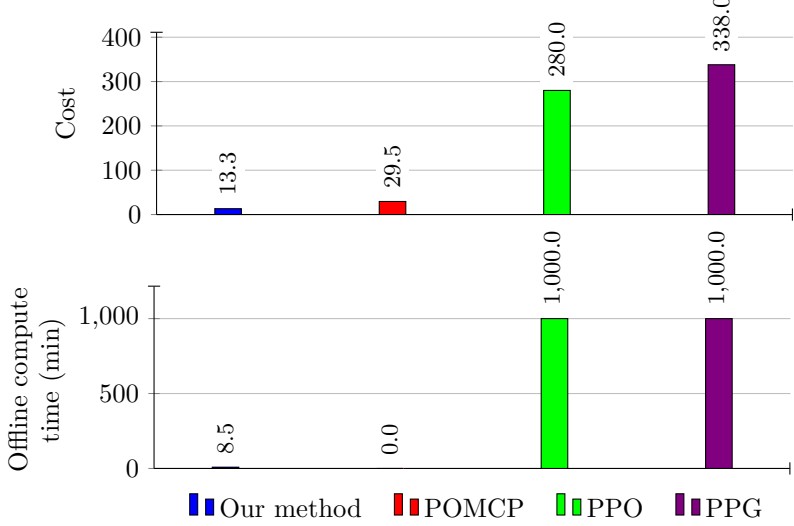

Figure 13: Evaluation results on the CAGE-2 POMDP. The upper plot shows the total discounted cost and the lower plot shows the (off-line) compute time in minutes. POMCP was given 30 seconds of computational budget per stage.

### 4.4 Memory Requirements

From a computational point of view, feature-based aggregation can reduce the required amount of computer memory compared to non-feature-based aggregation. In particular, storing a belief in memory for non-feature-based aggregation requires storing $n$ floating-point numbers. By contrast, storing a feature belief in

our method requires storing only $|\mathcal{F}|$ floating point numbers, where $|\mathcal{F}|$ denotes the number of feature states and $|\mathcal{F}| < n$. For example, in RS (10,10), we use 9,216 features instead of 102,401 states. As a consequence, the memory required to store a belief is reduced from about 400 KB to 36 KB (assuming a floating point number is represented by 32 bits).

Table 3 compares the memory required to store the cost function using our feature-based aggregation method with that of non-feature-based aggregation. To calculate the memory requirements, we assume that the cost for each belief point is represented by 32 bits and that our method is instantiated with the same features as used to compute the results in Figs. 10–12.

| POMDP Method | RS (4,4) | RS (5,5) | RS (5,7) | RS (7,8) | RS (10,10) | CAGE-2 |
|---|---|---|---|---|---|---|
| $\rho = 1$ | | | | | | |
| Feature-based aggregation | 1 KB | 3 KB | 12 KB | 49 KB | 36 KB | 1.6 MB |
| Non-feature-based aggregation | 1 KB | 3 KB | 12 KB | 49 KB | 400 KB | $4 \cdot 10^{23}$ YB |
| $\rho = 2$ | | | | | | |
| Feature-based aggregation | 128 KB | 1.2 MB | 19.5 MB | 300.1 MB | 0.2 GB | 340.5 GB |
| Non-feature-based aggregation | 128 KB | 1.2 MB | 19.5 MB | 300.1 MB | 19.5 GB | $2 \cdot 10^{70}$ YB |
| $\rho = 3$ | | | | | | |
| Feature-based aggregation | 10.7 MB | 326 MB | 20.4 GB | 1.2 TB | 0.5 TB | 46.28 PB |
| Non-feature-based aggregation | 10.7 MB | 326 MB | 20.4 GB | 1.2 TB | 650 TB | $6.5 \cdot 10^{116}$ YB |

Table 3: Required memory to store the cost function based on the features we used in our evaluation and assuming uniform discretization of the belief space using Eq. (9) with discretization resolution $\rho$.

More generally, our feature-based method allows us to control the memory consumption by tuning the feature space $\mathcal{F}$ and the number of representative feature beliefs $m$. In practice, we tune these parameters based on the available computing resources to achieve a suitable trade-off between cost and computational efficiency. Hence, our method does not necessarily use less memory than other computational POMDP methods but offers the flexibility to adapt the memory requirements to the computational resources and problem scale.

## 5 Biased Aggregation and the Approximation Errors

In this section, we introduce an extension of our method that leverages a known function $V$, which approximates the optimal cost function $J^*$. We refer to this extension as *biased aggregation*, to distinguish it from the *standard aggregation* approach described in the preceding sections. The biased aggregation approach was introduced in Bertsekas (2019a), and it is related to a classical DP scheme, known as "reward shaping" in the RL literature; see Ng et al. (1999). The biased aggregation can be viewed as the standard aggregation applied to a modified POMDP, obtained by incorporating bias terms into the stage costs of the original POMDP. These biases are defined using the given function $V$.

### 5.1 An Extension to Biased Aggregation

Suppose that we have a cost function approximation $V$ for a given POMDP, which maps $B$ to the real line $\Re$ and is bounded. We refer to this cost function approximation as the *bias function*. In the biased aggregation approach, we introduce a modified POMDP that is obtained from the original by changing its stage cost from $\hat{g}(b, u)$ to

$$\hat{g}(b, u) - V(b) + \alpha \sum_{z \in Z} \hat{p}(z \mid b, u) V\big(F(b, u, z)\big). \tag{49}$$

while the rest of the problem definition remains the same. Let us denote by $\tilde{V}$ the optimal cost function of the modified POMDP. It satisfies the corresponding Bellman equation

$$\tilde{V}(b) = \min_{u \in U} \left[ \hat{g}(b, u) - V(b) + \alpha \sum_{z \in Z} \hat{p}(z \,|\, b, u) \Big( \tilde{V}\big(F(b, u, z)\big) + V\big(F(b, u, z)\big) \Big) \right]$$

for all $b \in B$, or equivalently

$$\tilde{V}(b) + V(b) = \min_{u \in U} \left[ \hat{g}(b, u) + \alpha \sum_{z \in Z} \hat{p}(z \,|\, b, u) \Big( \tilde{V}\big(F(b, u, z)\big) + V\big(F(b, u, z)\big) \Big) \right]$$

for all $b \in B$. By comparing this equation with the Bellman equation for the original problem [cf. Eq. (4)], and in view of the fact that $J^*$ is the unique solution to Bellman's equation of the original problem, we see that the optimal cost functions of the modified and the original problems are related by

$$J^*(b) = \tilde{V}(b) + V(b), \qquad \text{for all } b \in B. \tag{50}$$

In particular, when $V(b) \approx J^*(b)$, we have $\tilde{V}(b) \approx 0$. Based on the error-bound analysis of Section 3. This suggests that biased aggregation is characterized by tighter error bounds than standard aggregation, and likely produces better approximations to $J^*$. Indeed, this is supported by our computational experiments.

Let us now apply our feature-based belief aggregation method to the modified problem. The controlled dynamic belief system constructed via our method is illustrated in Fig. 14. It can be seen that under the same policy and with the same sequence of random observations $z_k$, the modified and the original POMDP generate the same sequence of representative feature beliefs $\tilde{q}_k$ and beliefs $b_k$. Moreover, the associated Bellman equation can be characterized by an operator $\tilde{H}$ similar to that of the original problem, which maps $r \in \Re^m$ to $\tilde{H}r$ with components

$$(\tilde{H}r)(\tilde{q}) = \min_{u \in U} \left[ \hat{g}\big(D(\tilde{q}), u\big) + \alpha \sum_{z \in Z} \hat{p}\big(z \,|\, D(\tilde{q}), u\big) \left( V\Big(F\big(D(\tilde{q}), u, z\big)\Big) \right. \right.$$
$$\left. \left. + \sum_{\tilde{q}' \in \tilde{Q}} \psi_{G(\tilde{q}, u, z)\tilde{q}'} r_{\tilde{q}'} \right) \right] - V\big(D(\tilde{q})\big), \qquad \tilde{q} \in \tilde{Q}; \tag{51}$$

cf. Eq. (16).

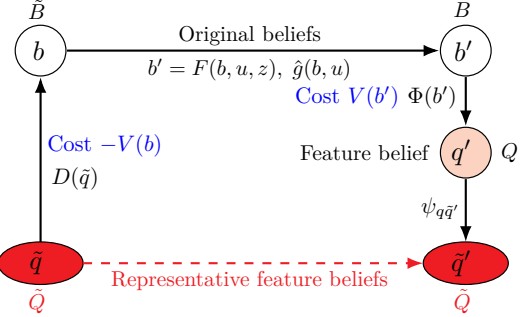

Figure 14: The controlled dynamic belief system constructed through our feature-based belief aggregation method for the modified POMDP with the bias function $V$; cf. Eq. (49).

Since $\tilde{H}$ is defined by simply modifying the stage costs in the operator $H$, it inherits the monotonicity and contractivity of $H$; see Prop. 1. Therefore, $\tilde{H}$ admits a unique fixed point $\tilde{r}^* = \tilde{H}\tilde{r}^*$ within $\Re^m$, which is the optimal cost of the aggregate problem with modified stage costs. Once $\tilde{r}^*$ is obtained, a cost function approximation $\tilde{J}$ of the original POMDP can be computed as

$$\tilde{J}(b) = V(b) + \sum_{\tilde{q} \in \tilde{Q}} \psi_{\Phi(b)\tilde{q}} \tilde{r}_{\tilde{q}}^*. \tag{52}$$

Moreover, from the definitions of $H$ and $\tilde{H}$, it can be seen that the VI algorithm and its asynchronous variant described in Section 2.3 can be modified accordingly for computing $\tilde{r}^*$. Compared with the standard aggregation approach, these algorithms do not involve excessive additional computations when solving for $\tilde{r}^*$. The only additional computation is evaluating $V$ at the beliefs $b$ that are also computed in the standard aggregation. Therefore, the computational savings discussed in Section 2.3, when compared with aggregation methods that operate directly on beliefs, remain valid in the biased aggregation approach.

We illustrate the biased aggregation method by applying it to our running example.

**Example 5.1.** *We apply biased aggregation to the problem instance described in Example 3.3 and compare it with standard aggregation. In particular, we consider the treasure hunting problem introduced in Example 1.1 with a single search site. The representative feature belief space $\tilde{Q}$ is constructed using the method described in Example 3.2 with varying discretization resolutions. The belief aggregation probabilities $\psi_{q\tilde{q}}$ are computed to satisfy the condition given in Eq. (42); cf. Example 3.3. Additional computational details are provided in Appendix A. We define the bias function $V$ to be the cost function approximation $\tilde{J}$ computed according to Eq. (18) using the unbiased aggregation method described in Example 3.3 with discretization resolution $\rho = 3$; cf. Fig. 15.*

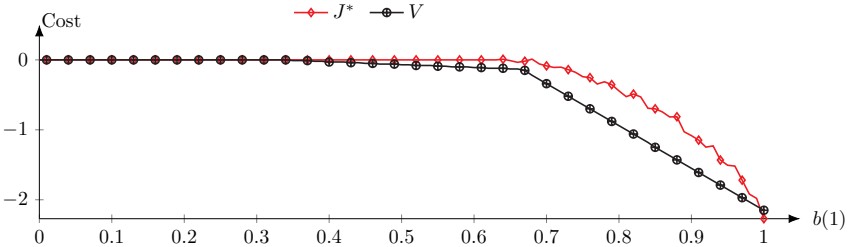

Figure 15: Comparison between the optimal cost function $J^*$ [cf. Eq. (4)] and the given cost function approximation $V$. In particular, the function $V$ is computed by using the standard aggregation approach with representative feature beliefs and the belief aggregation probabilities described in Example 3.3 with discretization resolution $\rho = 3$.

*In Fig. 16 we compare the cost function approximation $\tilde{J}$ for the biased and the unbiased case with varying discretization resolutions $\rho$ [cf. Eq. (9)]. We observe that the function $V$ biases the values of the cost function approximation $\tilde{J}$ to their correct levels. This bias allows us to obtain accurate cost function approximations with fewer representative feature beliefs than the unbiased case. In fact, we obtain a near-optimal approximation with only $m = 5$ representative feature beliefs when using biased aggregation.*

## 5.2 The Approximation Error

Let us now discuss briefly the approximation error of $\tilde{J}$; cf. Eq. (52). Since biased aggregation is standard aggregation applied to a modified POMDP, Props. 2, 3, and 4 readily apply to biased aggregation without modification. In addition, the counterparts of Props. 5 and 6 also hold by replacing $J^*$ with $J^* - V$, in view of the relation between $J^*$ and $\tilde{V}$ given in Eq. (50). Let us state without proofs these two results.

We first state the counterpart of Prop. 5 applied to the biased aggregation. For convenience, we recall that for every $\tilde{q} \in \tilde{Q}$, its footprint set $S_{\tilde{q}}$ is defined as

$$S_{\tilde{q}} = \{b \in B \mid \psi_{\Phi(b)\tilde{q}} > 0\}. \tag{53}$$

We have the following result.

**Proposition 7.** *The cost function approximation $\tilde{J}$ defined in Eq. (52) and the optimal cost function $J^*$ satisfy*

$$|\tilde{J}(b) - J^*(b)| \leq \frac{\epsilon}{1 - \alpha}, \qquad for\ all\ b \in B,$$

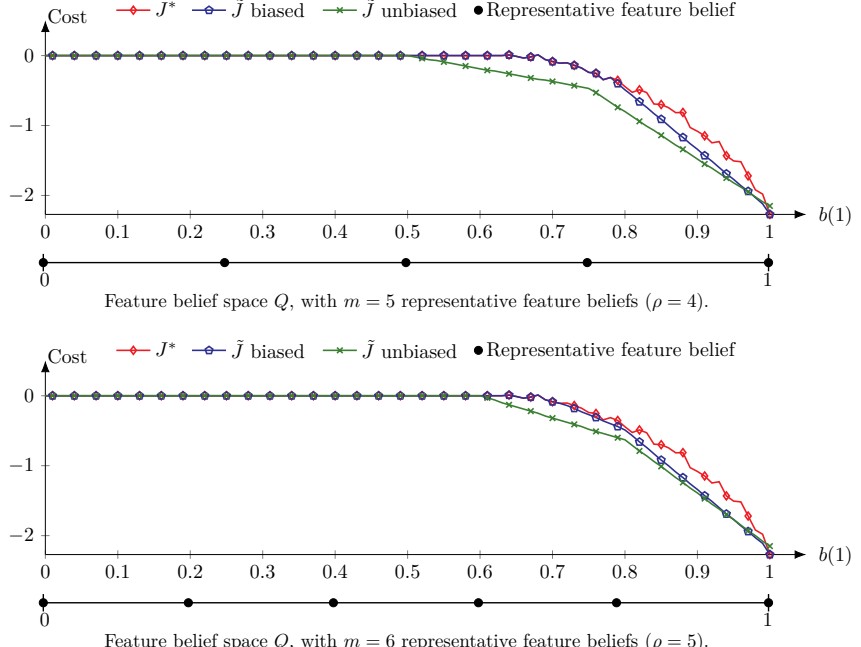

Figure 16: Comparison between the optimal cost function $J^*$ [cf. Eq. (4)] and the cost function approximations $\tilde{J}$ with and without the bias function $V$ for Example 5.1; cf. Eqs. (49) and (18). The belief aggregation probabilities $\psi_{q\tilde{q}}$ satisfy Eq. (42). The horizontal axes indicate the belief that the site has a treasure, and the vertical axes show the corresponding expected cost. The two plots here show the results where $\rho$ equals 4 and 5, respectively. This leads to a different number $m$ of representative feature beliefs, as indicated below the horizontal axes; cf. Eq. (9). In both cases, biased aggregation obtains a better approximation compared with standard aggregation.

*where $\epsilon$ is a finite constant defined by*

$$\epsilon = \max_{\tilde{q}\in\tilde{Q}} \sup_{b,b'\in S_{\tilde{q}}} |J^*(b) - V(b) - J^*(b') + V(b')|. \tag{54}$$

Finally, we state the counterpart of the error bound of Prop. 6 for biased aggregation, which requires that the convex hull of $\tilde{B}$ equals $B$, and the convex coefficient associated with $b \in B$ and $b' \in \tilde{B}$ is the belief aggregation probability $\psi_{\Phi(b)\Phi(b')}$; i.e.,

$$b = \sum_{b'\in\tilde{B}} \psi_{\Phi(b)\Phi(b')} b', \qquad \text{for all } b \in B. \tag{55}$$

Under this condition, we have the following result.

**Proposition 8.** *Let the disaggregation, aggregation, and belief aggregation probabilities satisfy Eq. (55). The cost function approximation $\tilde{J}$ defined in Eq. (52) and the optimal cost function $J^*$ satisfy*

$$0 \le J^*(b) - \tilde{J}(b) \le \frac{\epsilon}{1-\alpha}, \qquad \text{for all } b \in B,$$

*where $\epsilon$ is given by Eq. (54). In addition, if $\tilde{Q}$ contains $n$ representative feature beliefs, $\tilde{J} - V$ is linear.*

## 6    Conclusion

In this paper, we have introduced an aggregation-based method for approximating the optimal cost function of finite-state POMDP with infinite horizon and discounted costs. Our method leverages the classical

belief-space formulation by aggregating unobservable states into feature states, enabling the use of standard DP techniques to solve the resulting MDP. This approach reduces the necessary computation and enhances solution tractability. Through interpolation based on solutions to these MDP, we have obtained effective approximations of the optimal cost functions in the original POMDP. We have provided theoretical insights by establishing bounds on the approximation error, demonstrating the conditions under which our approximations form valid lower bounds for the optimal costs. Our method thus offers both practical computational advantages and meaningful theoretical guarantees in solving large-scale POMDP. We have also provided an extension of our method that biases the approximation towards the true optimal cost by incorporating a given cost function approximation.

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

# A Experimental Setup

We evaluate the discounted total cost for each POMDP based on 1000 simulation trials involving 100 stages. We run all simulations on an M2-ultra processor with the official POMDP parameters published in the literature.[5] For POMCP, PPO, and PPG, we use the parameters listed in Anonymous (2025c). For all other methods, we use the default hyperparameters in the available open-source implementation. In particular, for those methods where an open-source implementation is provided by the authors of the original paper, we use that implementation. For those methods where no open-source implementation is provided by the authors, we use the implementation and hyperparameters from the software library Egorov et al. (2017). For Lovejoy's method Lovejoy (1991), we use triangulation with the following granularities: 2 for RS (4,4), and 1 for the other POMDPs.

## A.1 Treasure Hunting

**Problem data**  The treasure values, search costs, and detection probabilities used in our running examples are as follows:

$$v = (6.48, 5.22, 5.43, 7.58, 3.09, 3.76, 8.01, 8.53, 7.86, 8.20),$$
$$c = (0.55, 0.86, 0.58, 0.86, 0.50, 0.98, 0.85, 0.74, 0.58, 0.84),$$
$$\beta = (0.13, 0.78, 0.11, 0.68, 0.11, 0.10, 0.30, 0.73, 0.54, 0.45).$$

These values correspond to a problem instance with 10 sites. For examples involving fewer than 10 sites ($N < 10$), we use the first $N$ entries of the vectors $v$, $c$, and $\beta$ to define the respective treasure values, costs, and detection probabilities. The discount factor $\alpha$ is 0.99.

**Feature-based belief aggregation**  In Examples 3.3 and 5.1, we require that the aggregation probabilities $\psi_{q\tilde{q}}$ satisfy the condition stated in Eq. (42) [or equivalently, Eq. (44)]. We design the belief aggregation probabilities such that each belief $b$ is represented by the convex hull of the "nearest" beliefs in $\tilde{B}$ based on the maximum norm; cf. Eq. (14).

**Off-line and on-line computation**  The aggregate problem is solved by VI given by Eq. (20). In both off-line and on-line computation, the belief estimator $F$ is computed exactly according to its definition; cf. Eq. (24).

## A.2 RS and CAGE-2

**Problem data**  The detailed problem data can be found in Anonymous (2025a).

---

[5]For the CAGE-2 POMDP, we use commit 9421c8e and the b-line attacker.

**Feature-based belief aggregation**  For RS (4,4)-RS (7,8), we associate each state $i$ with a unique feature state $x$; i.e., $X = \mathcal{F}$. For RS (10,10), we define a feature state $x$ as a 3-dimensional vector representing a node position in a $3 \times 3$ grid. We use $\phi_{ix}$ to aggregate the $10 \times 10$ grid positions of state $i$ into the positions of the $3 \times 3$ grid. This results in a feature space of cardinality $9,216$. In addition, we define $d_{xi}$ to be uniform across all states with matching features.

For CAGE-2, we define as a feature state $x$ a vector with three components: *attacker-state*, *attacker-target*, and *decoy-state*. The first two components represent the attacker's current location in the network and its next target node. The last component is the configuration of the deployed decoy services. We define $\phi_{ix}$ to map each state $i$ into $x$ uniformly. This results in a feature space of cardinality $427,500$. In addition, we define $d_{xi}$ to be uniform across all states with matching features.

For all problems, we define the set of representative feature beliefs $\tilde{Q}$ via uniform discretization using Eq. (9). We use discretization resolution $\rho = 3$ for RS (4,4); $\rho = 2$ for RS (5,5) and RS (5,7); and $\rho = 1$ for RS (7,8), RS (10,10), and CAGE-2. We associate these representative feature beliefs with feature beliefs $q \in Q$ via the nearest-neighbor mapping in Eq. (10).

**Off-line and on-line computation**  The aggregate problems are all solved by VI; cf. Eq. (20). For RS (4,4)-RS (7,8), the belief estimator $F$ is calculated exactly according to its definition [cf. Eq. (24)] in both off-line and on-line computation.

For RS (10,10) and CAGE-2, the belief estimator $F$ involved in both off-line and on-line computation is approximated by a rejection-sampling particle filter with 100 particles Crisan & Doucet (2002).

