# OpenReview forum: "Feature-Based Belief Aggregation for Partially Observable Markov Decision Problems"
_TMLR — Rejected by TMLR_

### Review · Reviewer_MFS4 · 2025-09-04

**Summary Of Contributions:**

The paper describes a belief aggregation technique for partially observable Markov decision processes.  The belief aggregation technique has two parts.  First the unobservable states are aggregated into a set of feature states.  Then beliefs over the feature states are discretized into a set of representative beliefs.  Overall, the dimensionality is reduced by the aggregation into a set of feature states and the infinite belief space is reduced to a finite set by the discretization.  This turns the POMDP into a fully observable MDP since we can treat the discrete beliefs as states.  The paper studies the approximation error introduced by the aggregation technique and provides bounds.  It also provides a limited empirical evaluation.

Strengths:
* Simple aggregation technique
* Clear and well written paper

Weaknesses:
* The work assumes that the transition function and observation function of the POMDP is known and therefore the work is applicable only to planning problems, which are rare in practice.
* The work lacks novelty.  The two phases of the aggregation technique are not new.  See below for more details.
* The empirical evaluation is limited to two artificial POMDPs
* The theoretical analysis provides a bound on the approximation error that may be vacuous. See below for more details.

**Additional Comments:**

None

**Audience:**

No

**Audience Explanation:**

The proposed aggregation technique is not new and is applicable only to planning problems where the transition and observation functions are known.

**Broader Impact Concerns:**

No concern

**Claims And Evidence:**

No

**Claims Explanation:**

The paper claims that the proposed aggregation technique is new, but the two phases are not new and their combination is not particularly novel.  The first phase consists of partitioning the unobservable states into clusters that are called feature states.  This idea has been around for quite some time and is often the first thing that anyone will try.  The paper does not propose a general technique to form the clusters.  Instead, it simply describes one heuristic tailored to a running example while referring the reader to Bertsekas’ book for general techniques, which confirms that state clustering is not new.  As for discretizing the beliefs into a finite set of beliefs, the paper simply proposes a uniform discretization scheme, which again is not new and leads to exponentially many beliefs (with respect to the number of feature states).  The paper does not dedicate any section to related work.  Had the authors done a survey of related work on state aggregation and belief discretization, they would have realized that the proposed techniques are not new.

The paper also claims that the proposed feature-based aggregation scheme generalizes existing aggregation methods.  I don’t see any generalization.  Partitioning states into clusters is not new.  Perhaps what the authors mean by “generalization” is that they are combining state aggregation with belief discretization, but combining two existing techniques does not constitute some kind of generalization.

The paper claims to establish a new bound on the approximation error of the cost function obtained through aggregation.  This bound $\epsilon/(1-\alpha)$ is provided in Proposition 5.  However, this bound may be vacuous. The worst possible approximation error for any POMDP can always be bounded by $(\max cost)/(1-\alpha)$.  In proposition 5, it is not clear whether epsilon is greater or smaller than max cost.  If it is greater than max cost, then the bound is vacuous.  The paper should provide conditions under which the bound will not be vacuous.

**Requested Changes:**

* The paper should include a related work section that clearly explains the differences and similarities with previous state aggregation techniques and belief discretization techniques.
* Since the proposed techniques are not new, the paper should also introduce new techniques.
* Conditions should be added to the error bound to ensure that it is not vacuous.

---

> ### Author Response · Authors · 2025-10-23
> **Rebuttal for Reviewer MFS4 Part 1**
>
> We thank the reviewer for the detailed feedback. The main concerns raised relate to (i) the assumption of known transition and observation models, (ii) the originality and scope of the proposed aggregation framework, (iii) the absence of a general clustering or discretization method, (iv) the interpretation of the error bound, and (v) the extent of the empirical evaluation. To address (i), we have clarified that the proposed algorithm can be implemented either using the explicit problem data or through a simulator. For (ii), we have expanded Section 2 to pinpoint the methodological extensions and we point out in Section 3 how our theoretical results go beyond existing ones. Regarding (iii), we note that developing a general clustering method is beyond the scope of this work. For (iv), we have explained in Section 3 that the derived approximation bound is intended not as a numerical guarantee but as an analytical tool for guiding feature and aggregation design, consistent with prior aggregation theory. Finally, we have expanded the experimental discussion to include additional examples such as Tiger, referenced through our public code repository.
>
> **(MFS4: 1)** *The work assumes that the transition function and observation function of the POMDP is known ... rare in practice.*
>
> **Authors' response:** We agree that our analysis uses transition and observation models, as is standard in theoretical studies of dynamic programming (DP) and approximate planning (e.g., Lovejoy 1991; Yu \& Bertsekas 2004; Saldi et al. 2017). One of our primary objectives is to analyze approximation properties and derive error bounds, which typically require a known model to define the Bellman operator precisely. That said, *the proposed method can be implemented either using the explicit problem data or through a simulator*, as noted in the paragraph after Eq.(23) of p.9: "the vector $b_k$ and the functions $\hat g$, $\hat{p}$, and $F$ can be written explicitly in closed form. Their values may also be estimated through simulation methods ..." and last paragraph in p.28: ``For RS (10,10) and CAGE-2, the belief estimator $F$ involved in both off-line and on-line computation is approximated by a rejection-sampling particle filter with $100$ particles Crisan \& Doucet (2002)." In other words, our approach applies equally when the model is given analytically, estimated from data, or accessed via simulation. We have expanded on these points in Sections 1 and 3 to clarify that the framework accommodates both model-based and simulation-based settings.
>
> In p.3, added the following paragraph:
> > Similar to other aggregation methods, the aggregate MDP obtained through our scheme can be solved either using explicit problem data or through simulation-based estimates. When the transition or observation probabilities are not available in closed form, the quantities required to construct and solve the aggregate MDP can be estimated through sampling from a simulator. This flexibility allows the proposed framework to be applied in both model-based and simulation-based contexts, and to remain effective even when only sample trajectories are accessible.
>
> In p.9, added the following sentence:
> > Hence, the same algorithmic structure applies whether explicit problem data or a simulator is available.
>
> **(MFS4: 2)** *The work lacks novelty. The two phases of the aggregation technique are not new. ... The paper claims that the proposed aggregation technique is new, but the two phases are not new and their combination is not particularly novel. The first phase consists of partitioning the unobservable states into clusters that are called feature states. This idea has been around for quite some time and is often the first thing that anyone will try.*
>
> **Authors' response:** We acknowledge that both state aggregation and belief discretization are well-established concepts. The contribution of our work lies not in redefining these operations individually, but in developing a hierarchical framework that integrates them in a mathematically consistent and analytically general form.
>
> Specifically, our construction introduces a mapping that aggregates unobservable states into feature states before belief formation, yielding a feature–belief space that remains Markovian and supports standard DP operations (Section 2.1, Fig. 2). The resulting framework unifies state aggregation (when beliefs are absent) and belief discretization (when $\mathcal F=X$) as special cases. (to be continued)

---

> ### Author Response · Authors · 2025-10-23
> **Rebuttal for Reviewer MFS4 Part 2**
>
> **Authors' response to (MFS4: 2):** (Continued) Importantly, our formulation is based on mild and natural yet necessary conditions [conditions stated in (a), (b), (c) and (d) Section 2.1] that ensure the resulting model is well-posed; omitting these can lead to pathological behaviors, as shown in Example 3.1. At the same time, the proposed framework relaxes the classical disjoint-partition requirement (the sets $S_{\tilde q}$ need not be disjoint), thereby generalizing both the theoretical results and the structural framework found in prior aggregation literature. We added the following paragraph at the end of Section 2.1:
>
> > Note that our framework encompasses existing aggregation methods as special cases. In particular, when each state $i$ can be identified exactly from the observation, the formulation reduces to the aggregation framework for perfect state information problems described in (Bertsekas, 2012, Section 6.5). Conversely, by setting $\mathcal F=X$, we recover the aggregation method applied directly to the belief space, as discussed in (Bertsekas, 2012, Example 6.5.5). Even within these contexts, our framework goes further by allowing more flexible choices of aggregation and disaggregation probabilities while maintaining theoretical guarantees, as will be discussed in Section 3.2. In addition, our framework addresses the special challenges posed by large-scale POMDP problems, for which there are few (if any) theoretical or computational studies in the context of aggregation (and no studies in the case of biased aggregation, as we will discuss in Section 5).
>
> **(MFS4: 3)** *The paper does not propose a general technique to form the clusters. Instead, it simply describes one heuristic tailored to a running example while referring the reader to Bertsekas’ book for general techniques, which confirms that state clustering is not new.*
>
> **Authors' response** This observation is correct, and it reflects an intentional design choice. Our framework is meant to be general with respect to the choice of features, not to prescribe a particular clustering algorithm. This is consistent with the aggregation literature (Bertsekas 2012, Ch. 6), where the aim is to analyze approximation behavior for any admissible feature mapping rather than to optimize over clustering heuristics. In the revision, we have made this explicit by adding the following paragraph in p.5.
>
> > The feature states $x$ and associated sets $I_x$, along with the aggregation and disaggregation probabilities, can be designed using problem-specific structure and engineering intuition. Alternatively, suitable feature states can be derived automatically through deep learning techniques. For instance, given a simulator of the POMDP and a policy, one can generate a dataset of belief–cost pairs and train a deep neural network to approximate the policy’s cost function. The learned representation can then serve as a basis for feature-state system design; see Bertsekas (2018) for a detailed discussion for perfect-state-information problems. We do not prescribe a specific way of constructing feature states $x$ and associated sets $I_x$ in our method, as the appropriate approach depends on the problem domain. Instead, we allow arbitrary choices of feature states (as long as the sets $I_x$ are disjoint), consistent with the classical aggregation theory framework.
>
> **(MFS4: 4)** *As for discretizing the beliefs into a finite set of beliefs, the paper simply proposes a uniform discretization scheme, which again is not new and leads to exponentially many beliefs (with respect to the number of feature states).*
>
> **Authors' response** *Our framework applies to cases beyond uniform discretization over the feature belief space.* We use uniform discretization in the experiments as a neutral baseline, since the focus of the paper is the theoretical structure rather than discretization design. Other discretization schemes—such as adaptive grids or sample-based representative beliefs—can be used within the same framework without affecting the underlying analysis.
> Regarding scalability, while a full uniform grid would grow exponentially, in practice only a small subset of reachable beliefs is encountered through simulation-based aggregation. We have clarified the flexibility of the discretizations of the feature beliefs in Section 2.1.
>
> In p.6, added the following sentence:
>
> > Our framework places no restriction on how the set $\tilde Q$ is constructed, allowing any discretization of the feature-belief space that is convenient for the problem at hand.

---

> ### Author Response · Authors · 2025-10-23
> **Rebuttal for Reviewer MFS4 Part 3**
>
> **(MFS4: 5)** *The paper does not dedicate any section to related work. Had the authors done a survey of related work on state aggregation and belief discretization, they would have realized that the proposed techniques are not new.*
>
> **Authors' response:** We appreciate this comment. The introduction already provides a focused and balanced discussion of the relevant literature, including Lovejoy (1991), Tsitsiklis \& Van Roy (1996), Yu \& Bertsekas (2004), Roy et al. (2005), and Saldi et al. (2017). These works each consider either state aggregation or belief discretization under more restrictive assumptions—typically requiring disjoint partitioning of the underlying space.
>
> Our framework builds directly upon and generalizes these studies by (i) allowing non-disjoint footprint sets $S_{\tilde q}$, and (ii) introducing a feature layer that defines beliefs over aggregated states, thus connecting state aggregation and belief approximation under a single unified structure. The construction rests on natural but essential regularity conditions to ensure well-posedness and to avoid pathological cases such as those illustrated in Example 3.1.
>
> Therefore, while a separate related-work section is not necessary, the current introduction already situates our contribution clearly within—and as an extension of—the existing aggregation literature.
>
> Also see our response to (MFS4: 2) and (MFS4: 6).
>
> **(MFS4: 6)** *The paper also claims that the proposed feature-based aggregation scheme generalizes existing aggregation methods. I don’t see any generalization. Partitioning states into clusters is not new. Perhaps what the authors mean by “generalization” is that they are combining state aggregation with belief discretization, but combining two existing techniques does not constitute some kind of generalization.*
>
> **Authors' response:** By *generalization* we mean a formal extension that covers established aggregation paradigms as special cases while relaxing classical assumptions. Concretely:
> - *POMDP case*: With $\mathcal F=X$, our formulation recovers standard MDP aggregation framework applied directly to the belief state formulation of POMDP, as discussed in (Bertsekas 2012, Ch. 6).
> - *Fully observable case*: When the underlying problem is fully observable (MDP with perfect state information), applying the same framework reduces to classical MDP state aggregation.
> - *Relaxed assumptions*: Our construction does not require the footprint sets $S_{\tilde q}$ to be disjoint, unlike prior treatments. This overlap allowance of the sets $S_{\tilde q}$ enables more flexible choices of aggregation probabilities, belief aggregation probabilities, and underpins the more general approximation relation (Proposition 5), as discussed in the second paragraph in p.15.
>
> These properties hold under natural but necessary regularity conditions that preclude pathologies (see Example 3.1). Thus, the generalization is not a mere combination of known steps; it is a broader, rigorously defined framework that subsumes prior cases and extends their theoretical guarantees. In addition, our method addresses the special challenges posed by large-scale POMDP problems, for which there are few (if any) theoretical or computational studies in the context of aggregation (and no studies in the case of biased aggregation).
>
> Also see our response to (MFS4: 2).
>
> **(MFS4: 7)** *The empirical evaluation is limited to two artificial POMDPs*
>
> **Authors' response:** The empirical study includes five RockSample benchmarks and the CAGE-2 environment. In addition, we have tested the method on standard small-scale problems such as Tiger, where our method performs on par with other state-of-the-art aggregation and approximation techniques. Because such problems are well understood and offer little additional insight, we have only mentioned them briefly in the revision and referenced detailed results in our public GitHub repository.
>
> We summarize the results of Tiger as the following footnote in p. 18:
>
> > We also evaluated the performance of our method on the classical Tiger problem studied in Kaelbling et al. (1998). Our method obtains the optimal cost function when using a discretization resolution of $\rho=100$. Comparable performance (in terms of both cost and computation time) is achieved by PBVI, SARSOP, POMCP, HSVI, and AdaOps. Other methods either yield suboptimal solutions or require substantially longer computation times to reach optimality.
>
> (to be continued)

---

> ### Author Response · Authors · 2025-10-23
> **Rebuttal for Reviewer MFS4 Part 4**
>
> **Authors' response to (MFS4: 7):** (Continued) The CAGE-2 environment, by contrast, models a realistic server system with dynamic load allocation, comprising more than $10^{47}$ latent states. This domain demonstrates the practical scalability and real-world relevance of the proposed framework, as it represents a class of high-dimensional decision-making problems where belief aggregation yields substantial computational advantages.
> We have emphasized these points in Section 4 of the revised manuscript and refer to experimental evaluation on a real testbed reported in other works that {validate} our claim here. Also see our response to (1HAz: 3).
>
> **(MFS4: 8)** *The theoretical analysis provides a bound on the approximation error that may be vacuous.*
>
> **Authors' response:** We thank the reviewer for raising this concern. The bound in Proposition 5 is intended as an analytical characterization, not as a computable numerical estimate. Its primary role is to isolate how the approximation error depends on the aggregation structure, in particular, the variation of the optimal cost $J^*$ within each footprint set $S_{\tilde q}$, and on the contraction factor $\alpha$. This decomposition clarifies the sources of approximation error and provides design guidance for constructing effective feature mappings and aggregation probabilities.
>
> This interpretive role is consistent with the classical aggregation literature (Bertsekas \& Tsitsiklis 1996; Yu \& Bertsekas 2004), where bounds of this type are not meant to be numerically tight but to convey structural insight. Importantly, our formulation is established under natural yet necessary conditions that ensure the model is well-posed; omitting these can lead to pathological behaviors (Example 3.1). At the same time, it generalizes existing theoretical results by allowing overlapping footprint sets $S_{\tilde q}$, thus extending the scope of prior error analyses.
>
> We have clarified in the text (following Prop. 5) that the bound serves as a conceptual tool to guide the choice of features and aggregation mappings rather than as a direct quantitative guarantee.
>
> In p.15, we added:
>
> > Consistent with the aggregation literature Tsitsiklis \& van Roy (1996), (Bertsekas, 2012, Section 6.5), and Bertsekas (2018), the bound is not intended as a computable numerical quantity but as an analytical characterization of how the approximation error depends on the aggregation structure, specifically, on the variation of $J^\star$ within each footprint set. This interpretation clarifies the source of approximation error and provides guidance for effective choices of design parameters in our method. In particular, we seek a feature space $\mathcal{F}$, disaggregation probabilities $d_{xj}$, and aggregation probabilities $\phi_{jx}$ so that the optimal cost $J^*$ is approximately constant over the sets $S_{\tilde{q}}$. The bound also relies on mild, natural yet necessary conditions [(a)–(d) in Section 2.1] that ensure the model is well posed; omitting these can lead to pathological cases such as those illustrated in Example~3.1.

---

### Review · Reviewer_1HAz · 2025-09-14

**Summary Of Contributions:**

The authors propose a method for solving POMDP with intractably large state spaces. The method relies on well-designed feature states, over which a belief space is maintained; moreover, the space of feature beliefs is discretized into representative beliefs, over which an MDP is synthesized. Solving this MDP (e.g. through dynamics programming) results in a cost function which may then be mapped back to the POMDP, and used as an estimate. The authors present a bound on the approximation error of this scheme, which depends on the difference in value between pairs of beliefs with the same representation, and study conditions under which the cost estimate is a lower bound on the optimal cost. The method is then empirically evaluated on CAGE-2 and a selection of Rocksample problems, and compares favorably to a wide number of alternative approaches for solving the POMDP, particularly in terms of compute. The results are then extended to a biased aggregation approach, which results in tighter error bounds.

**Additional Comments:**

- The choice of features will inevitably affect the gap between estimated and real cost. Would it be possible to connect the bound in Proposition 5 to the quality of features? In my opinion the impact of this paper would benefit if the approximation error itself could be used to guide the design of features directly.
- The hyperlink for Proposition 6 seems to point to Figure 6 instead.
- The performance of PPO and PPG seems to be identical on RS. Is this the case? If so, why?

**Audience:**

Yes

**Audience Explanation:**

I am not familiar with this line of work, so my evaluation in this regard does not carry a lot of confidence. While the idea in itself is simple, I am not aware of similar works. Since learning over features and abstracted states is in practice very effective, for instance in Deep RL, I suspect that this paper's results might also be interesting outside of this field.

**Claims And Evidence:**

Yes

**Claims Explanation:**

The paper is clear and very well written, which is helpful for following through the description of the method and its support (running examples and Figure 2 are particularly helpful). The main contribution of this work is the introduction of an algorithm, and the main claim regard its soundness and performance. To the best of my knowledge, the algorithm is sound, and the guarantees provided are not too surprising, but correct, although my evaluation is limited by my lack of familiarity with the literature. The evaluation is also overall convincing, despite only involving two problems.

**Requested Changes:**

- Although the authors mention they can be learned, the main drawback of the method appears to be in the design of features. This limitation is not acknowledged sufficiently in my opinion. Would it be possible to introduce a detailed discussion of how hard this design process can be, and of which environments might be more or less suited to it?
- I suggest adding a clear discussion on why the two current benchmarks are selected.
- I would ask the authors to consider whether the method could be evaluated on additional problems (the simplest option would be the treasure-seeking problem presented as a motivating example).
- Are there existing feature-based approaches for solving POMDPs (e.g. that described on the 7th line of page 3)? Are they tractable in this setting and, if so, can they be included in the empirical evaluation?
- I would suggest relocating proofs, as well as the last section, to the appendix. The text is rather long at the moment.
- Could the authors include a formal definition on "one-to-one" (page 10)?

---

> ### Author Response · Authors · 2025-10-23
> **Response to Reviewer 1HAz Part 1**
>
> We thank the reviewer for the encouraging and constructive comments. The main concerns relate to a) adding a discussion about techniques for selecting features; and b) adding specific technical details and motivations for the experimental evaluation. We agree and have included both.
>
> **(1HAz: 1)**: *The main drawback of the method appears to be in the design of features. (...) Would it be possible to introduce a detailed discussion of how hard this design process can be?*
>
> **Authors' response:** We agree that the design of features is an important aspect when applying our method in practice. The design of features can leverage problem-specific insights and intuition. We provide three examples of this in our experimental evaluation, where we design features for the Treasure hunting, RockSample and the CAGE-2 problems. Moreover, our method can be combined with deep learning techniques for automated feature extraction.
>
> We added this paragraph to p.5:
>
> > The feature states $x$ and associated sets $I_x$, along with the aggregation and disaggregation probabilities, can be designed using problem-specific structure and engineering intuition. Alternatively, suitable feature states can be derived automatically through deep learning techniques. For instance, given a simulator of the POMDP and a policy, one can generate a dataset of belief–cost pairs and train a deep neural network to approximate the policy’s cost function. The learned representation can then serve as a basis for feature-state aggregation; see Bertsekas (2018) for a detailed discussion for perfect-state-information problems. We do not prescribe a specific way of constructing feature states $x$ and associated sets $I_x$ in our method, as the appropriate approach depends on the problem domain. Instead, we allow arbitrary choices of feature states (as long as the sets $I_x$ are disjoint), consistent with the classical aggregation theory framework.
>
> **(1HAz: 2)**: *I suggest adding a clear discussion on why the two current benchmarks are selected.*
>
> **Authors' response:** We are happy to provide such a discussion. In short, we select RockSample because it is the standard benchmark problem in the POMDP literature, and we select CAGE-2 because it is a new problem of unprecedented scale. Moreover, CAGE-2 is based on a real system, with experimental validation results reported elsewhere.
>
> We added this paragraph to p.8:
>
> > We chose these two problems for the following reasons. The RS problem serves as the de facto standard benchmark in the POMDP literature and allows us to compare our method with other methods proposed in prior research. In contrast, CAGE-2 is a more recent and substantially larger-scale POMDP that captures the complexity of realistic decision-making environments. Specifically, CAGE-2 is derived from a use case involving the operation and defense of a networked system against cyberattacks, which is an active topic of investigation in the cybersecurity research community. As a consequence, CAGE-2 enables us to assess the applicability of our method to a problem with high practical relevance.
>
>
> **(1HAz: 3)** *I would ask the authors to consider whether the method could be evaluated on additional problems (the simplest option would be the treasure-seeking problem presented as a motivating example).*
>
> **Authors' response:** In principle, our method is applicable to any POMDP. We have already applied our method to the treasure-hunting problem, as shown in Figs. 5--9. We chose this problem as a running example because it has a structure that allows for the efficient computation of the optimal solution, which enables us to illustrate our theoretical results and measure the approximation error.
>
> Upon the reviewer's suggestion, we have also applied our method to the Tiger POMDP from (Kaelbling, Littman, and Cassandra, 1998). The results are shown in the table below. We used discretization resolution $\rho=100$ to instantiate our method.
>
>
> | Method      | Cost        | Compute time (min) |
> |--------------|-------------|--------------------|
> | Ours         | $-19.37$    | $10^{-5}$ |
> | PBVI         | $-19.37$    | $10^{-3}$ |
> | SARSOP       | $-19.37$    | $10^{-5}$ |
> | POMCP        | $-19.37$    | $1.6$ |
> | HSVI         | $-19.37$    | $10^{-5}$ |
> | PPO          | $-19.37$    | $2$ |
> | QMDP         | $-18.64$    | $10^{-3}$ |
> | PPG          | $-19.37$    | $2$ |
> | AdaOPS       | $-19.37$    | $1.6$ |
> | R-DESPOT     | $-18.17$    | $1.6$ |
> | Roy et al.   | $-14.12$    | $10^{-2}$ |
> | POMCPOW      | $-14.17$    | $1.6$ |
> | Lovejoy      | $-17.56$    | $10^{-2}$ |
>
> **Caption:**
> Evaluation results on the Tiger POMDP.
>
> (To be continued)

---

> ### Author Response · Authors · 2025-10-23
> **Response to Reviewer 1HAz Part 2**
>
> **Authors' response to (1HAz: 3):** (Continued) The testing results on Tiger are summarized as the following footnote in p.18:
>
> > We also evaluated the performance of our method on the classical Tiger problem. Our method obtains the optimal cost function when using a discretization resolution of $\rho=100$. Comparable performance (in terms of both cost and computation time) is achieved by PBVI, SARSOP, POMCP, HSVI, and AdaOps. Other methods either yield suboptimal solutions or require substantially longer computation times to reach optimality.
>
> **(1HAz: 4)** *Are there existing feature-based approaches for solving POMDP (e.g., that described on the 7th line of page 3)? Are they tractable in this setting and, if so, can they be included in the empirical evaluation?*
>
> **Authors' response:** To the best of our knowledge, our approach is the first feature-based aggregation method tailored for POMDP. The methods referenced on page 3 are feature-based methods for perfect-state-information problems, namely stadard MDP. One can apply this aggregation approach directly to the belief space formulation of a POMDP. However, it is only feasible for small scale problem. When the number of hidden states $n$ is large, this approach becomes computationally intractable, both due to the memory requirement and also the computational demands for updating the beliefs. For example, as detailed in Appendix A.2, in RS (4,4)-RS (7,8), we associate each state $i$ with a unique feature state $x$; i.e., $X=\mathcal{F}$ they do not apply directly to POMDP. In this case, our method is standard aggregation for perfect state information problems applied directly to POMDP. However, this becomes infeasible for larger problems, such as RS (10,10) and CAGE-2. See the newly added Section 4.4 for the memory requirements. Also see our response to (TPEP: 3).
>
> **(1HAz: 5)** *I would suggest relocating proofs, as well as the last section, to the appendix. The text is rather long at the moment.*
>
> **Authors' response:** We thank the reviewer for the suggestion. Ultimately, the placement of proofs depends on style and tradition. Since this is a theoretical paper, we think it is useful to have the proofs stated inline with the theoretical results so that the interested reader can follow the reasoning chronologically. We deliberately put the extension to biased aggregation in a separate section so that the reader can distinguish the extension from the core method. Since this extension is one of our contributions, we think it belongs to the main body rather than the appendix.
>
> **(1HAz: 6)** *Could the authors include a formal definition on ``one-to-one" (page 10)?*
>
> **Authors' response:** Thank you for the suggestion.
>
> We have added the following footnote in p.10:
>
> > We say a function $f$ that maps the set $\tilde Q$ to the set $\tilde B$ is one-to-one, if i) for every $\tilde b$, there exists some $\tilde q\in \tilde Q$ such that $f(\tilde q)=\tilde b$, and ii) $f(\tilde q)\neq f(\tilde q')$ for every $\tilde q,\tilde q\in \tilde Q$ and $\tilde q\neq \tilde q'$.
>
> **(1HAz: 7)** *Would it be possible to connect the bound in Proposition 5 to the quality of features? In my opinion, the impact of this paper would be benefited if the approximation error itself could be used to guide the design of features directly.*
>
> **Authors' response:** We thank the reviewer for this suggestion. In short, Prop. 5 implies that to have low approximation error, we seek features that lead to ``footprint sets" $S_{\tilde{q}}$ over which the optimal cost function is approximately constant.
>
> We added this paragraph to p. 15:
>
> > Consistent with the aggregation literature Tsitsiklis \& van Roy (1996), (Bertsekas, 2012, Section 6.5), and Bertsekas (2018), the bound is not intended as a computable numerical quantity but as an analytical characterization of how the approximation error depends on the aggregation structure, specifically, on the variation of $J^{*}$ within each footprint set. This interpretation clarifies the source of approximation error and provides guidance for effective choices of design parameters in our method. In particular, we seek a feature space $\mathcal{F}$, disaggregation probabilities $d_{xj}$, and aggregation probabilities $\phi_{jx}$ so that the optimal cost function is approximately constant over the sets $S_{\tilde{q}}$. The bound also relies on mild, natural yet necessary conditions [(a)–(d) in Section 2.1] that ensure the model is well posed; omitting these can lead to pathological cases such as those illustrated in Example 3.1.
>
> Please note that computing the approximation error requires knowledge of the optimal cost function. Hence, while the approximation error provides qualitative insight, it is not always feasible to compute in practice.

---

> ### Author Response · Authors · 2025-10-23
> **Response to Reviewer 1HAz Part 3**
>
> **(1HAz: 8)** *The hyperlink for Proposition 6 seems to point to Figure 6 instead.*
>
> **Authors' response:** Thanks for notifying us about this potential issue. We checked our paper, and the hyperlink pointing to Figure 6 appears twice in the paper (p. 14 and p. 15), both of which are correct. We did not find the incorrect link.
>
> **(1HAz: 9)** *The performance of PPO and PPG seems to be identical on RS. Is this the case? If so, why?*
>
> **Authors' response:** In our experiments on the RockSample problem, PPO and PPG converge to similar policies. We believe that the reason for the similar policies are a) there are only a few reasonable policies for the RockSample problem; and b) both PPO and PPG are policy gradient methods.
>
> We added this paragraph to p. 20:
>
> > In our experiments, both PPO and PPG converge to the same suboptimal policy in the RS problem. This is likely because both methods are policy-gradient algorithms that optimize parameterized policies through gradient-based updates, which can cause them to become trapped in similar local optima. Since we observe that PPO and PPG converge to different policies in the CAGE-2 problem (as detailed in the next subsection), another explanation may be that the RS problem admits only a few reasonable policies, limiting the diversity of converged solutions.

---

> > ### Comment · Reviewer_1HAz · 2025-10-23
> >
> > I would like to thank the authors for their clarification and the additional experiment on the Tiger POMDP. I have no further questions.

---

### Review · Reviewer_TPEP · 2025-10-15

**Summary Of Contributions:**

This paper presents a belief aggregation approach to POMDP solving. It computes a POMDP policy by identifying a much smaller belief space, namely feature belief space, and then framing the POMDP solving as solving MDP in the feature belief space. A feature belief space is a belief (probability distribution) over a feature space, which is manually selected from the partially observed states of the POMDP problem. Key to the approach is in inferring actions in the much smaller feature belief space, but then apply this action and its value computation in the original POMDP belief space. This computation is done at every step, but only the feature belief is maintained, the original belief is not maintained but computed as needed, based on the feature belief, which in turn significantly reduce memory requirements, and has the potential to approximate the value function better than prior belief aggregation methods. This computation is possible due to the one-to-one disaggregation function that maps feature beliefs to original POMDP beliefs, and aggregation function that does the opposite. A thorough analysis is provided. Numerical experiments, together with comparisons with standard comparators in POMDP planning, are provided.

**Audience:**

Yes

**Audience Explanation:**

This paper will be of interest to people working in planning and learning under uncertainty, esp. those working on MDP and POMDP.

**Claims And Evidence:**

Yes

**Claims Explanation:**

The proposed approach can be beneficial in improving the scalability of POMDP solving, and the paper is quite thorough. There has been work on simplifying belief space in POMDP solving, such as [Roy, et.al. 2005] and Joelle Pineau, Geoffrey Gordon, and Sebastian Thrun, Policy-contingent abstraction for robust robot control, UAI 2012. However, the proposed aggregation method is novel and supported by a much more thorough analysis than many prior work in this line of approach.

**Requested Changes:**

The benefit of this approach relies heavily on the feature selected. It would be beneficial to understand how these features affect performance, or at least provide some guidelines on how to select good features beyond they have to be disjoint. Please provide elaboration on this aspect.

The reference for SARSOP in Table 2 is incorrect. The correct ref for SARSOP is Hanna Kurniawati, David Hsu, and Wee Sun Lee, Sarsop: Efficient Point-based POMDP planning by approximating optimally reachable belief spaces, Proc. Robotics: Science and systems, 2008. The provided reference [Ong, et.al. 2010] is for MOMDP, which is an efficient POMDP factorization when the problem consists of fully and partially observed state variables. The Open Source code APPL (Offline) provides SARSOP with and SARSOP without MOMDP factorization.

Please provide the memory consumption in the experimental results. Given memory is one of the main scalability issue with POMDP solving, and an aim of the proposed method is to alleviate this issue, showing the memory consumption could show significant practical benefit of the proposed approached compare to existing solvers.

---

> ### Author Response · Authors · 2025-10-23
> **Response to Reviewer TPEP Part 1**
>
> We thank the reviewer for the positive and constructive feedback. The main suggestions by the reviewer are a) to include a discussion on techniques for designing the feature states; and b) to discuss the memory requirements of our method. We agree with these suggestions and have revised the manuscript to address both.
>
> **(TPEP: 1)** *The benefit of this approach relies heavily on the feature selected. It would be beneficial to understand how these features affect performance, or at least provide some guidelines on how to select good features beyond they have to be disjoint. Please provide elaboration on this aspect.*
>
> **Authors' response:** Thank you for the suggestion. We are happy to provide such guidelines. In particular, we have clarified how features can be derived based on problem-specific structure or be automatically learned through deep learning techniques. We have also clarified how Prop. 5 provides qualitative insight into the design of features.
>
> We added this paragraph to p.5:
>
> > The feature states $x$ and associated sets $I_x$, along with the aggregation and disaggregation probabilities, can be designed using problem-specific structure and engineering intuition. Alternatively, suitable feature states can be derived automatically through deep learning techniques. For instance, given a simulator of the POMDP and a policy, one can generate a dataset of belief–cost pairs and train a deep neural network to approximate the policy’s cost function. The learned representation can then serve as a basis for feature-state aggregation; see Bertsekas (2018) for a detailed discussion for perfect-state-information problems. We do not prescribe a specific way of constructing feature states $x$ and associated sets $I_x$ in our method, as the appropriate approach depends on the problem domain. Instead, we allow arbitrary choices of feature states (as long as the sets $I_x$ are disjoint), consistent with the classical aggregation theory framework.
>
> We added this paragraph to p. 15:
>
> > Consistent with the aggregation literature Tsitsiklis \& van Roy (1996), (Bertsekas, 2012, Section 6.5), and Bertsekas (2018), the bound is not intended as a computable numerical quantity but as an analytical characterization of how the approximation error depends on the aggregation structure, specifically, on the variation of $J^{*}$ within each footprint set. This interpretation clarifies the source of approximation error and provides guidance for effective choices of design parameters in our method. In particular, we seek a feature space $\mathcal{F}$, disaggregation probabilities $d_{xj}$, and aggregation probabilities $\phi_{jx}$ so that the optimal cost function is approximately constant over the sets $S_{\tilde{q}}$. The bound also relies on mild, natural yet necessary conditions [(a)–(d) in Section 2.1] that ensure the model is well posed; omitting these can lead to pathological cases such as those illustrated in Example 3.1.
>
> **(TPEP: 2)**: *The reference for SARSOP in Table 2 is incorrect. The correct ref for SARSOP is Hanna Kurniawati, David Hsu, and Wee Sun Lee, Sarsop: Efficient Point-based POMDP planning by approximating optimally reachable belief spaces, Proc. Robotics: Science and systems, 2008.*
>
> **Authors' response:** Thank you for notifying us about this. We have corrected the reference.

---

> ### Author Response · Authors · 2025-10-23
> **Response to Reviewer TPEP Part 2**
>
> **(TPEP: 3)**: *Please provide the memory consumption in the experimental results.*
>
> **Authors' response:** The reviewer is correct that the feature-based approach allows to control the memory consumption.
>
> We have added section 4.4 to the paper with the following text:
>
> > From a computational point of view, feature-based aggregation can reduce the required amount of computer memory compared to non-feature-based aggregation. In particular, storing a belief in memory for non-feature-based aggregation requires storing $n$ floating-point numbers. By contrast, storing a feature belief in our method requires storing only $|\mathcal{F}|$ floating point numbers, where $|\mathcal{F}|$ denotes the number of feature states and $|\mathcal{F}| < n$. For example, in RS (10,10), we use 9,216 features instead of 102,401 states. As a consequence, the memory required to store a belief is reduced from about 400 KB to 36 KB (assuming a floating point number is represented by 32 bits).
>
> > Table 3 compares the memory required to store the cost function using our feature-based aggregation method with that of non-feature-based aggregation. To calculate the memory requirements, we assume that the cost for each belief point is represented by $32$ bits and that our method is instantiated with the same features
> as used to compute the results in Figs. 10–12.
>
> > More generally, our feature-based method allows us to control the memory consumption by tuning the feature space $\mathcal{F}$ and the number of representative feature beliefs $m$. In practice, we tune these parameters based on the available computing resources to achieve a suitable trade-off between cost and computational efficiency. Hence, our method does not necessarily use less memory than other computational POMDP methods but offers the flexibility to adapt the memory requirements to the computational resources and problem scale.
>
> ### Table: Required memory to store the cost function
>
> | Method \\ POMDP | RS (4,4) | RS (5,5) | RS (5,7) | RS (7,8) | RS (10,10) | CAGE-2 |
> |------------------|-----------|-----------|-----------|-----------|-------------|---------|
> | **$\rho = 1$** |  |  |  |  |  |  |
> | Feature-based aggregation | 1 KB | 3 KB | 12 KB | 49 KB | 36 KB | 1.6 MB |
> | Non-feature-based aggregation | 1 KB | 3 KB | 12 KB | 49 KB | 400 KB | 4·10²³ YB |
> | **$\rho = 2$** |  |  |  |  |  |  |
> | Feature-based aggregation | 128 KB | 1.2 MB | 19.5 MB | 300.1 MB | 0.2 GB | 340.5 GB |
> | Non-feature-based aggregation | 128 KB | 1.2 MB | 19.5 MB | 300.1 MB | 19.5 GB | 2·10⁷⁰ YB |
> | **$\rho = 3$** |  |  |  |  |  |  |
> | Feature-based aggregation | 10.7 MB | 326 MB | 20.4 GB | 1.2 TB | 0.5 TB | 46.28 PB |
> | Non-feature-based aggregation | 10.7 MB | 326 MB | 20.4 GB | 1.2 TB | 650 TB | 6.5·10¹¹⁶ YB |
>
> **Caption:**
> Required memory to store the cost function based on the features used in our evaluation and assuming uniform discretization of the belief space with discretization resolution $\rho$.

---

### Decision · Action_Editor_MRsK · 2025-12-11

**Recommendation:** Reject

**Audience:**

No

**Audience Explanation:**

This is a bit harder to judge.  I think there are definitely findings in this paper that some individuals in TMLR's audience would be interested in knowing.  And that is basically the question being asked in TMLR's criteria.  However, it's not true of all of the paper's findings.  Some of the findings are not properly contextualized (whether that's the nuance of what's new, or how one can and should interpret the newly introduced approximation bound).  I think saying one thing that's of interest, in a long paper saying many things is not the intent of this TMLR criterion.

However, I think that judging the concerns around evidentiary support, or rather stating those claims with appropriate clarity and nuance, would largely address this criterion as well.

**Claims And Evidence:**

No

**Claims Explanation:**

There is not an easy to reach decision for this paper.  The reviewers final recommendations are quite disparate, including recommendations of accept, reject, and revise.  I am not fully persuaded by Reviewer MFS4, but I do think they offer a significant critique that is unaddressed by the author response and mild revision.  This critique ultimately questions if the claims in the submission are convincingly supported, and it certainly has persuaded two of the three reviewers to find the paper not meeting the TMLR guidelines in its current form.

First, let me say that I'm not persuaded by the critique that the method is limited to settings where the transition and observation probabilities are known.  The author response seemed to interpret this critique as requiring a closed form for these functions (and have addressed in changes that no such closed form is needed as simulations are sufficient or provide estimates).  However, I think the reviewer's intended critique is that the work is limited to POMDP planning, rather than attempting to learn observation/transition functions through interaction.  This indeed is a limitation of this work.  However, I think POMDP planning still remains an interesting problem in its own right.  Similarly, I don't think the choice of "artificial POMDPs" somehow detracts from the paper's advance.

As for the other critiques, I do think the authors claims of a "new belief aggregation" method are not convincingly justified.  This message of a new method are not presented with any sort of nuance in the paper, but they are at best true with significant nuance.  This nuance includes a general framework for how such aggregations can be constructed that can be used to describe past approaches (what the authors appear to mean by "generalization"), as well as lifting a potentially significant restriction on aggregation resulting in disjoint footprint sets.  I think if the claims of contribution of the paper were made more carefully, e.g.,  by noting this lifted restriction, it would be more clearly justified.  However, then this would raise the question of whether there are justifiable benefits from this lifted restriction or broader generalization, which would be a focal point of the paper.  Note that this is not judging the paper on lack of novelty, but rather judging the paper on its claims.

Similarly, I am sympathetic to the reviewer's critique of the possibly vacuous approximation bound.  The authors defense is not convincing.  If the bound is so loose as to be vacuous, then the terms and shape of the bound cannot be trusted to be useful.  For example, I could present an arbitrarily loose bound where performance depends on the day of the week.  The proof the bound doesn't suddenly justify that the algorithm's performance might depend on the day of the week.

I think a significant rewrite and shift in focus of the contribution and claims of the paper could make it meet the evidence requirement.  But by focusing the contributions it likely also changes the nature of evidence needed to justify the newly stated claims.

**Resubmission Of Major Revision:**

The authors may consider submitting a major revision at a later time.